# IBCL: Zero-shot Model Generation under Stability-Plasticity Trade-offs

## Abstract

Algorithms that balance the stability-plasticity trade-off are well-studied in the continual learning literature. However, only a few of them focus on obtaining models for specified trade-off preferences. When solving the problem of continual learning under specific trade-offs (CLuST), state-of-the-art techniques leverage rehearsal-based learning, which requires retraining when a model corresponding to a new trade-off preference is requested. This is inefficient since there exist infinitely many different trade-offs, and a large number of models may be requested. As a response, we propose Imprecise Bayesian Continual Learning (IBCL), an algorithm that tackles CLuST efficiently. IBCL replaces retraining with constant-time convex combination. Given a new task, IBCL (1) updates the knowledge base in the form of a convex hull of model parameter distributions and (2) generates one Pareto-optimal model per given trade-off via convex combination without any additional training. That is, obtaining models corresponding to specified trade-offs via IBCL is zero-shot. Experiments whose baselines are current CLuST algorithms show that IBCL improves by at most 45% on average per task accuracy and by 43% on peak per task accuracy, while maintaining a near-zero to positive backward transfer. Moreover, its training overhead, measured by number of batch updates, remains constant at every task, regardless of the number of preferences requested. Details at: `https://github.com/ibcl-anon/ibcl`.

## 1 Introduction

Continual learning (CL), also known as lifelong machine learning, is a special case of multi-task learning, where tasks arrive in temporal sequence one-by-one (Chen and Liu, 2016; Parisi et al., 2019; Ruvolo and Eaton, 2013b; Thrun, 1998). Two key properties matter for CL algorithms: stability and plasticity (De Lange et al., 2021). Here, stability means the ability to maintain performance on previous tasks, not forgetting what the model has learned, and plasticity means the ability to adapt to a new task. Unfortunately, these two properties are conflicting due to the multi-objective optimization nature of CL (Kendall et al., 2018; Sener and Koltun, 2018). For years, researchers have been balancing the stability-plasticity trade-off. However, few have discussed the problem of learning models for specifically given trade-off points. In this paper, we focus on such a problem, which we denote as CL under specific trade-offs (CLuST).

Why is CLuST important? First, in certain scenarios, it is important to explicitly specify *how much* stability and plasticity are needed, in order to obtain one customized model per trade-off preference. Second, when there exist a large number of preferences, the *efficiency* of training every customized model matters. Otherwise, the training cost accumulates over all preferences and becomes prohibitive. Therefore, we not only seek a solution for CLuST, but also an efficient one.

Let us consider an example of a movie recommendation system. The model is first trained to rate movies in the sci-fi genre. Then, a new genre, e.g., documentaries, is added by the movie company. The model needs to learn how to rate documentaries while not forgetting how to rate sci-fis. Training this model boils down to a CL problem. The company now wants to build a recommendation system that adapts to users' tastes in movies. For example, Alice has equal preferences over sci-fis and documentaries. Bob, however, wants to watch only documentaries and has no interest in sci-fis at all. Consequently, the company aims to train two customized models for Alice and Bob, respectively, to predict how likely a sci-fi or a documentary is to be recommended. Based on individual

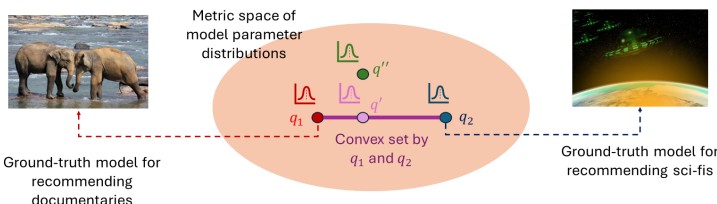

Figure 1: A Bayesian view of a Pareto-optimal distribution $q'$ and a non-Pareto-optimal distribution $q''$.

preferences, Alice's personal model should balance between the accuracy in rating sci-fis and rating documentaries, while Bob's model allows to compromise the accuracy in rating sci-fis to achieve a high accuracy in rating documentaries. As new genres are added, users should be able to input their preferences over all available genres to obtain customized models. Since there could be a large number of different users, and each user's taste in movie genres would vary over time, the movie company should implement models that adapt to a potentially infinite number of preferences. The cost would be prohibitive if the company had to train one model per distinct preference.

To formalize the CLuST problem, we take a Bayesian perspective, where learnable model parameters are viewed as random variables (Farquhar and Gal, 2019; Kessler et al., 2023; Nguyen et al., 2018). As illustrated in Figure 1, we consider all parameter distributions living in a metric space. This metric can be any valid metric for distributions, such as the 2-Wasserstein distance (Deza and Deza, 2013). The figure shows an example of two tasks, with their ground-truth distributions being $q_1$ and $q_2$, respectively. From this setup, a distribution emphasizing stability (at task 2) is a distribution closer to $q_1$ than $q_2$, and one prioritizing plasticity is closer to $q_2$ than $q_1$. Notice that irrespective of the desired stability-plasticity trade-off, we want the distribution to be *Pareto-optimal*, which loosely means that there is no way to improve such a distribution by making it closer to *both* $q_1$ and $q_2$. We can see that Pareto-optimality is equivalent to being inside the convex set enclosed by $q_1$ and $q_2$. For example, $q'$ in the figure is a Pareto-optimal distribution, while $q''$ is not. With this setting, we can specify a trade-off point using a *preference vector* (Mahapatra and Rajan, 2020; 2021) $\bar{w} = (w_1, w_2)$, where $w_1, w_2 \geq 0$ and $w_1 + w_2 = 1$. The preferred Pareto-optimal distribution is therefore a convex combination $w_1 q_1 + w_2 q_2$.

So far, researchers have already proposed the use of preference vectors to specify trade-off points in multi-task and continual learning (Gupta et al., 2021; Lin et al., 2019; 2020; Ma et al., 2020). However, instead of using them as coefficients for convex combinations, state-of-the-art techniques use them as regularizers in *rehearsal-based methods*. That is, existing algorithms that aim to solve CLuST memorize some data $d_i$ for each task $i$ (for "rehearsal"), and let the loss at task $i$ be $l_i = \sum_{j=1}^{i} w_j l(d_j)$, with $l$ being a generic loss function like cross-entropy. There are at least two drawbacks to this approach. First, rehearsals have to retrain the entire model whenever we have a new trade-off preference. **In plain words, these methods have a training overhead proportional to the number of preferences at each task.** As there exist infinitely many possible preferences, this boils down to an efficiency issue when there is a large number of preferences, such as a large number of users in the movie recommendation example. It would be desirable if, instead of retraining, we could obtain the preferred models by constant-time, training-free operations. Moreover, rehearsals have to cache data, and stable performance on previous tasks depends on which data can be memorized.

To overcome these shortcomings faced by CLuST algorithms, we propose Imprecise Bayesian Continual Learning (IBCL), whose workflow is illustrated in Figure 2. At step 1, upon the arrival of a new task's training data, IBCL update its *knowledge base* (that is, all information shared across tasks) in the form of a convex set of distributions with finitely many extreme elements (the elements that cannot be written as convex combinations of one another), called *finitely generated credal set* (FGCS) (Caprio et al., 2024). This is done by variational inference from the previous task's learned distribution, and the learned distributions serve as extreme elements of the FGCS. Each point in the FGCS corresponds to one Pareto-optimal distribution on the trade-off polytope of all tasks so far. Then, at step 2, given any preference vector $\bar{w}$, IBCL selects the preferred distribution by convex combination. A parameter region is obtained as a highest density region (HDR) of the distribu-

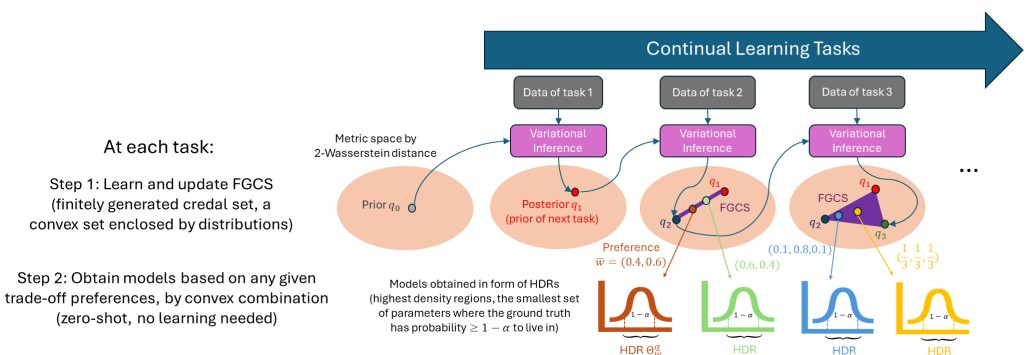

Figure 2: The workflow of Imprecise Bayesian Continual Learning (IBCL). Here, we start from 1 prior, but in practice there may be more than 1 to reduce epistemic uncertainty.

tion selected, which is the smallest parameter set that contains the ground-truth model with high probability.

IBCL tackles the identified shortcomings as follows. **First, IBCL replaces retraining in state-of-the-art with constant-time, zero-shot convex combination to generate models. It has constant training overhead per task (to update the FGCS), independent of the number of preferences.** In addition, no cache of data is required, and therefore the stability of our model does not depend on the data memorized. Experiments on image classification and NLP benchmarks support the effectiveness of IBCL. We find that IBCL improves on baselines by at most 45% in average per task accuracy and by 43% in peak per task accuracy, while maintaining a near-zero to positive backward transfer, with a constant training overhead regardless of number of preferences. We also show that IBCL has a sublinear memory growth along the number of tasks.

Overall, we have the following contributions: (1) We are the first to rigorously formulate the CLuST problem, which asks for efficiency upon a large number of preferences (Section 3). (2) We propose IBCL, a Bayesian CL algorithm to solve the CLuST problem (Section 4). (3) We experiment on standard image classification and NLP benchmarks to support our claims (Section 5).

## 2 BACKGROUND

Our algorithm hinges upon the concepts of finitely generated credal set (FGCS) from Imprecise Probability (IP) theory (Augustin et al., 2014; Caprio et al., 2024; Walley, 1991).

**Definition 1** (Finitely Generated Credal Set). Given a finite set of probability distributions $\{q^j\}_{j=1}^m$, a finitely generated credal set (FGCS) is the convex set

$$\mathcal{Q} = \left\{ q : q = \sum_{j=1}^m \beta^j q^j, \forall j\, \beta^j \geq 0, \sum_{j=1}^m \beta^j = 1 \right\}. \tag{1}$$

In other words, FGCS $\mathcal{Q}$ is the convex hull $\text{CH}(\{q^j\}_{j=1}^m)$ of finitely many distributions $\{q^j\}_{j=1}^m$. That is, given a finite collection of distributions $\{q^j\}_{j=1}^m$ (that we call the extreme elements of the credal sets, and denote by $\text{ex}[\mathcal{Q}]$), $\mathcal{Q}$ is the collection of all probability distributions $q$ that can be written as a convex combination of the $q^j$'s. If the state space is finite, then the $q^j$'s can be seen as probability vectors, whose entries represent the probability mass assigned by distribution $q^j$ to the elements of the state space.

Next, we borrow the idea of highest density region (HDR) (Coolen, 1992).

**Definition 2** (Highest Density Region). Let $\theta \in \Theta$ be a continuous random variable following a probability density function (pdf) $q$, with $\Theta$ being a set of interest. Given a significance level $\alpha \in [0, 1]$, the $(1 - \alpha)$-HDR is a subset $\Theta_q^\alpha \subset \Theta$, such that

$$\int_{\Theta_q^\alpha} q(\theta) d\theta \geq 1 - \alpha, \text{ and } \int_{\Theta_q^\alpha} d\theta \text{ is minimal.} \tag{2}$$

In equation 2, requiring that $\int_{\Theta_q^\alpha} d\theta$ is minimal corresponds to requiring that $\Theta_q^\alpha$ has the smallest possible cardinality (i.e. the least possible number of elements). Indeed, if $\Theta$ is finite, it can be replaced by "$|\Theta_q^\alpha|$ is minimal". Since we consider the most general case (in which set $\Theta$ may be uncountable), we must use the integral notion in place of cardinality, as pointed out, e.g., in Coolen (1992). In turn, equation 2 tells us that $\Theta_q^\alpha$ is the set having the smallest number of elements that also satisfies $\Pr_{\theta \sim q}[\theta \in \Theta_q^\alpha] \geq 1 - \alpha$, provided that $\theta \sim q$. A detailed explanation with an illustrated example of Definition 2 is in Appendix B.

We are also interested in Bayesian Continual learning (BCL) (Nguyen et al., 2018), as follows.

**Definition 3** (Bayesian Continual Learning). BCL is a class of CL procedures, which starts with a prior distribution $q_0$. At a task $i \in \{1, 2, ...\}$, we are given i.i.d. training data $\{(x_s, y_s)\}_{s=1}^{n_i} \subset \mathcal{X} \times \mathcal{Y}$ of inputs and outputs (the $x$'s and $y$'s, respectively). The Bayesian model is updated from prior distribution $q_{i-1}$ to posterior $q_i$ using the labeled data. Then, $q_i$ used as a prior in task $i + 1$.

Other related works, including multi-task and continual learning, are reviewed in Appendix C.

## 3 FORMULATING THE CLuST PROBLEM

In this section we formalize the CLuST problem. We consider domain-incremental learning (Van de Ven and Tolias, 2019) for classification models, with an unbounded number of stability-plasticity trade-off preferences at each task. The goal is to construct a learning algorithm with training overhead independent of the number of preferences, and that enjoys performance guarantees.

### 3.1 ASSUMPTIONS

Let $\mathcal{X}$ be the space of inputs, and $\mathcal{Y}$ be the space of labels. In a typical classification problem, $\mathcal{X}$ will be a subset of a Euclidean space, and $\mathcal{Y}$ a finite set. In a typical regression problem, $\mathcal{Y}$ will too be a subset of a Euclidean space. In general, we do not limit ourselves to either scenario. As a consequence, we let the input and the output spaces be generic sets. Call $\Delta_{\mathcal{X}\mathcal{Y}}$ the space of all possible distributions on $\mathcal{X} \times \mathcal{Y}$. A task $i$ is associated with a distribution $p_i \in \Delta_{\mathcal{X}\mathcal{Y}}$, from which labeled data can be i.i.d. drawn. We assume that all tasks are similar to each other.

**Assumption 1** (Task Similarity). *For all task $i$, $p_i \in \mathcal{F}$, where $\mathcal{F}$ is a convex subset of $\Delta_{\mathcal{X}\mathcal{Y}}$. Also, we assume that the diameter of $\mathcal{F}$ is some $r > 0$, that is, $\sup_{p,q \in \mathcal{F}} \|p - q\|_{W_2} \leq r$, where $\| \cdot \|_{W_2}$ denotes the 2-Wasserstein distance.*

The reason for choosing the 2-Wasserstein metric is that it enables easy computation of convex combination of distributions. Under this metric, the weighted sum of distributions can be derived from a weighted sum of means and standard deviations. Details are in Appendix D.

Notice that Assumption 1 does not tell us that all tasks have the same distribution, but merely that the true data generating processes pertaining to different tasks are not too distant from one another. In addition, such a notion of "being not too distant" is entirely in the hands of the user, via the choice of radius $r$ and of the metric to endow $\Delta_{\mathcal{X}\mathcal{Y}}$. This is rather natural: for the time being, we do not expect e.g. a robot to be able to fold our clothes (task 1), and then deliver a payload in a combat zone (task 2). Assumption 1 is needed to mitigate the possible model misspecification, which in turn could lead to catastrophic forgetting even when Bayesian inference is carried out exactly ((Kessler et al., 2023) and Appendix E).

More formally, under Assumption 1, for any two tasks $i$ and $j$, their underlying distributions $p_i$ and $p_j$ are "close enough", i.e. $\|p_i - p_j\|_{W_2} \leq r$. Moreover, since $\mathcal{F}$ is convex, any convex combination of task distributions belongs to $\mathcal{F}$. Next, we assume the parameterization of class $\mathcal{F}$.

**Assumption 2** (Parameterization of Task Distributions). *Every distribution $F$ in $\mathcal{F}$ is parameterized by $\theta$, a parameter belonging to a parameter space $\Theta$.*

An example of a parameterized family that satisfies Assumption 1 is given in Appendix F. Notice that all tasks share the same input space $\mathcal{X}$ and label space $\mathcal{Y}$, and we do not have task ids as an additional input, so the learning is domain-incremental (Van de Ven and Tolias, 2019). We then formalize stability-plasticity trade-off preferences over tasks.

**Definition 4** (Stability-plasticity Trade-off Preferences over Tasks). *Consider $k$ tasks with underlying distributions $p_1, p_2, \ldots, p_k$. We express a stability-plasticity trade-off preference (or simply, a preference) over them via a probability vector $\bar{w} = (w_1, w_2, \ldots, w_k)^\top$, that is, $w_i \geq 0$ for all $i \in \{1, \ldots, k\}$, and $\sum_{i=1}^k w_i = 1$.*

Based on this definition, given a preference $\bar{w}$ over all $k$ tasks encountered, the personalized model for the user aims to learn the distribution $p_{\bar{w}} := \sum_{i=1}^k w_i p_i$. It is the distribution associated with tasks $1, \ldots, k$ that also takes into account a preference over them. Since $p_{\bar{w}}$ is the convex combination of $p_1, \ldots, p_k$, thanks to Assumptions 1 and 2, we have that $p_{\bar{w}} \in \mathcal{F}$, and therefore it is also parameterized by some $\theta \in \Theta$.

The learning procedure is Bayesian domain-incremental learning. That is, the learning follows BCL as in Definition 3, and all data and label distributions are similar, as per Assumption 1, without any knowledge of task id's. Then, at any task $k$, we are given at least one user preference $\bar{w}$ over the $k$ tasks so far. The data drawn for task $k + 1$ will not be available until we have finished learning models for all preferences at task $k$.

## 3.2 MAIN PROBLEM

We aim to design a domain-incremental learning algorithm that generates one model per preference over tasks , with unbounded number of preferences and tasks. Given a significance level $\alpha \in [0, 1]$, at any task $k$, the algorithm should satisfy:

1. **Zero-shot preferred model generation**. Let $\bar{w}$ be a preference over the $k$ tasks. When there are more than one preference $\bar{w}_s$, $s \in \{1, 2, \ldots\}$, no training is needed for generating models, for all $s > 1$. That is, the model generation for new preferences is zero-shot.

2. **Probabilistic Pareto-optimality**. Let $\hat{q}_{\bar{w}}$ denote the convex combination of the estimated parameter distributions for tasks $1, \ldots, k$ using preference weights $\bar{w}$. We want to identify the subset of model parameters having the least amount of members, $\Theta^\alpha_{\hat{q}_{\bar{w}}} \subset \Theta$ (written as $\Theta^\alpha_{\bar{w}}$ for notational convenience from now on), that the Pareto-optimal parameter $\theta^\star_{\bar{w}}$ (i.e. the ground-truth parameter of $p_{\bar{w}}$) belongs to with high probability, i.e., $\Pr_{\theta^\star_{\bar{w}} \sim \hat{q}_{\bar{w}}}[\theta^\star_{\bar{w}} \in \Theta^\alpha_{\bar{w}}] \geq 1 - \alpha$.

3. **Sublinear buffer growth**. The memory overhead for the entire procedure should be growing sublinearly in the number of tasks.

## 4 IMPRECISE BAYESIAN CONTINUAL LEARNING

As shown in Figure 2, IBCL performs two steps at each task. First, it updates a knowledge base in form of an FGCS (Section 4.1). Second, it uses a convex combination of the extreme elements of the FGCS, instead of retraining, to zero-shot generate models under given preferences (Section 4.2).

## 4.1 FGCS KNOWLEDGE BASE UPDATE

As discussed in the Introduction, we take a Bayesian Continual Learning (BCL) approach, that is, the parameter $\theta$ of distribution $p_k$ pertaining to task $k$ is viewed as a random variable distributed according to some distribution $q$.

At the beginning of the analysis, we specify $m$ many such distributions, $\text{ex}[\mathcal{Q}_0] = \{q_0^1, \ldots, q_0^m\}$. They are the ones that the designer deems plausible – a priori – for parameter $\theta$ of task 1. Upon observing data pertaining to task 1, we learn a set $\mathcal{Q}_1^{tmp}$ of posterior parameter distributions and buffer them as extreme elements $\text{ex}[\mathcal{Q}_1]$ of the FGCS $\mathcal{Q}_1$ corresponding to task 1. We proceed similarly for the successive tasks $i \geq 2$.

In Algorithm 1, at task $i$, we approximate $m$ posteriors $q_i^1, \ldots q_i^m$ via variational inference from buffered priors $q_{i-1}^1, \ldots q_{i-1}^m$ one-by-one (line 3). Variational inference is a standard Bayesian learning procedure that minimizes the evidence lower bound (ELBO) loss to infer a posterior distribution from a prior and observed data (Nguyen et al., 2018). However, we do not want to buffer all learned posteriors, so we use a distance threshold $d$ to exclude the posteriors that are similar to the distributions that are already buffered (lines 4 - 10). When distributions similar to $q_i^j$ (within threshold

---

**Algorithm 1** FGCS Knowledge Base Update

**Input**: Current knowledge base in the form of FGCS extreme elements $\text{ex}[\mathcal{Q}_{i-1}] = \{q_{i-1}^1, \ldots, q_{i-1}^m\}$, observed labeled data $(\bar{x}_i, \bar{y}_i) = \{(x_{1_i}, y_{1_i}), \ldots, (x_{n_i}, y_{n_i})\}$ at task $i$, and distribution distance threshold $d \geq 0$

**Output**: Updated extreme elements $\text{ex}[\mathcal{Q}_i]$

1: $\mathcal{Q}_i^{tmp} \leftarrow \emptyset$
2: **for** $j \in \{1, \ldots, m\}$ **do**
3:     $q_i^j \leftarrow \mathsf{variational\_inference}(q_{i-1}^j, \bar{x}_i, \bar{y}_i)$
4:     $d_i^j \leftarrow \min_{q \in \text{ex}[\mathcal{Q}_{i-1}]} \|q_i^j - q\|_{W_2}$
5:     **if** $d_i^j \geq d$ **then**
6:         $\mathcal{Q}_i^{tmp} \leftarrow \mathcal{Q}_i^{tmp} \cup \{q_i^j\}$                    ▷ Store distribution $q_i^j$
7:     **else**
8:         $q_i^j \leftarrow \arg\min_{q \in \text{ex}[\mathcal{Q}_{i-1}]} \|q_i^j - q\|_{W_2}$     ▷ Fetch the stored distribution with minimal
   distance to $q_i^j$, and overwrite $q_i^j$ with a pointer to that distribution
9:         $\mathcal{Q}_i^{tmp} \leftarrow \mathcal{Q}_i^{tmp} \cup \{q_i^j\}$                    ▷ Only a pointer is stored
10:     **end if**
11: **end for**
12: $\text{ex}[\mathcal{Q}_i] \leftarrow \text{ex}[\mathcal{Q}_{i-1}] \cup \mathcal{Q}_i^{tmp}$

---

$d$) are found in the knowledge base, we store a pointer to the distribution with minimal distance in place of $q_i^j$, and do not memorize $q_i^j$ (lines 8-9). The posteriors that are sufficiently different from the already buffered distributions are then appended to the knowledge base (line 12).

Notice that the memory overhead of Algorithm 1 is remembering at most $m$ distributions into $\mathcal{Q}_i^{tmp}$ at line 6. In practice, $m$ is a small constant (we choose $m = 3$ in experiments). Therefore, the memory complexity is $O(1)$. Moreover, some newly memorized distributions may be discarded and replaced by a previous distribution in cache at line 8. With larger threshold $d$ at line 5, more distributions are discarded at lines 8-9. The amortized memory complexity analysis under different threshold $d$'s is discussed in our ablation studies, see Section 5.2.

The time complexity of Algorithm 1 is dominated by variational inference at line 3. Every variational inference costs a non-negligible training time, which we denote as $O(v)$. There are a total of $m$ variational inferences computed at each task. Since $m$ is a constant, the overall time complexity is still $O(v)$.

## 4.2 Zero-shot Generation of User Preferred Models

Next, after having updated the FGCS extreme elements for task $i$, we are given a set of user preferences. For each preference $\bar{w}$, we need to identify the Pareto-optimal parameter $\theta_{\bar{w}}^\star$ for the preferred data distribution $p_{\bar{w}}$. This procedure can be divided into two steps as follows.

First, we find the parameter distribution $\hat{q}_{\bar{w}}$ via a convex combination of the extreme elements in the knowledge base, whose weights correspond to the entries of preference vector $\bar{w}$. That is,

$$\hat{q}_{\bar{w}} = \sum_{k=1}^i \sum_{j=1}^{m_k} \beta_k^j q_k^j \text{ where } \sum_{j=1}^{m_k} \beta_k^j = w_k, \text{ and } \beta_k^j \geq 0, \text{ for all } j \text{ and all } k. \tag{3}$$

Here, $q_k^j$ is a buffered extreme point of FGCS $\mathcal{Q}_k$, i.e. the $j$-th parameter posterior of task $k$. The weight $\beta_k^j$ of this extreme point is decided by preference vector entry $\bar{w}_j$. In implementation, if we have $m_k$ extreme elements stored for task $k$, we can choose equal weights $\beta_k^1 = \cdots = \beta_k^m = w_k/m_k$. For example, if we have preference $\bar{w} = (0.8, 0.2)^\top$ on two tasks so far, and we have two extreme elements per task stored in the knowledge base, we can use $\beta_1^1 = \beta_1^2 = 0.8/2 = 0.4$ and $\beta_2^1 = \beta_2^2 = 0.2/2 = 0.1$.

As we can see from the following theorem, distribution $\hat{q}_{\bar{w}}$ is a parameter posterior corresponding to a preference elicitation via preference vector $\bar{w}$ over the tasks encountered so far.

**Theorem 1** (Selection Equivalence). *Selecting a precise distribution $\hat{q}_{\bar{w}}$ from $\mathcal{Q}_i$ is equivalent to specifying a preference weight vector $\bar{w}$ on $p_1, \ldots, p_i$.*

Please refer to Appendix G for the proof. Theorem 1 entails that the selection of $\hat{q}_{\bar{w}}$ in Algorithm 2 is related to the correct parameterization of $p_{\bar{w}} \in \Delta_{\mathcal{X}\mathcal{Y}}$.

Second, we compute the HDR $\Theta_{\bar{w}}^\alpha \subset \Theta$ from $\hat{q}_{\bar{w}}$. This is implemented via a standard procedure that locates the smallest region in the parameter space whose enclosed probability mass is (at least) $1 - \alpha$, according to $\hat{q}_{\bar{w}}$. This procedure can be routinely implemented, e.g., in R, using package HDInterval (Juat et al., 2022). As a result, we locate the smallest set of parameters $\Theta_{\bar{w}}^\alpha \subset \Theta$ associated with the preference $\bar{w}$. This subroutine is formalized in Algorithm 2. Notice that this computation is simply a convex combination, i.e., a weighted sum of all distributions in $\text{ex}[\mathcal{Q}_i]$. In practice, the summation is defined under 2-Wasserstein metric, which takes constant time to compute as detailed in Appendix D. Therefore, the time complexity is O(1). Also, this algorithm does not produce any memory overhead.

---

**Algorithm 2** Preference HDR Computation

---

**Input**: Knowledge base $\text{ex}[\mathcal{Q}_i]$ with $m_k$ extreme elements saved for task $k \in \{1, \ldots, i\}$, preference $\bar{w}$ on the $i$ tasks, significance level $\alpha \in [0, 1]$
**Output**: HDR $\Theta_{\bar{w}}^\alpha \subset \Theta$

1: **for** $k = 1, \ldots, i$ **do**
2: $\quad \beta_k^1 = \cdots = \beta_k^m \leftarrow w_k / m_k$
3: **end for**
4: $\hat{q}_{\bar{w}} = \sum_{k=1}^{i} \sum_{j=1}^{m_k} \beta_k^j q_k^j$
5: $\Theta_{\bar{w}}^\alpha \leftarrow \text{hdr}(\hat{q}_{\bar{w}}, \alpha)$

---

### 4.3 Overall IBCL Algorithm and Analysis

From the two subroutines in Sections 4.1 and 4.2, we construct the overall IBCL algorithm as in Algorithm 3.

---

**Algorithm 3** Imprecise Bayesian Continual Learning

---

**Input**: Prior distributions $\text{ex}[\mathcal{Q}_0] = \{q_0^1, \ldots, q_0^m\}$, hyperparameters $\alpha$ and $d$
**Output**: HDR $\Theta_{\bar{w}}^\alpha$ for each given preference $\bar{w}$ at each task $i$

1: **for** task $i = 1, 2, \ldots$ **do**
2: $\quad \bar{x}_i, \bar{y}_i \leftarrow$ sample $n_i$ labeled data points i.i.d. from $p_i$
3: $\quad \text{ex}[\mathcal{Q}_i] \leftarrow \text{fgcs\_update}(\text{ex}[\mathcal{Q}_{i-1}], \bar{x}_i, \bar{y}_i, d)$          ▷ % Algorithm 1 %
4: $\quad$ **while** user has a new preference **do**
5: $\quad\quad \bar{w} \leftarrow$ user input
6: $\quad\quad \Theta_{\bar{w}}^\alpha \leftarrow \text{preference\_hdr\_comput}(\text{ex}[\mathcal{Q}_i], \bar{w}, \alpha)$      ▷ % Algorithm 2 %
7: $\quad$ **end while**
8: **end for**

---

For each task, in line 3, we use Algorithm 1 to update the knowledge base by learning $m$ posteriors from the current priors. In lines 5-7, according to a user-given preference over all tasks so far, we obtain the HDR of the model associated with preference $\bar{w}$ in zero-shot via Algorithm 2. Notice that this HDR computation does not require the initial priors $\text{ex}[\mathcal{Q}_0]$, so we can discard them once the posteriors $\mathcal{Q}_1$ are learned in the first task.

The overall time complexity is dominated by the $O(v)$ variational inference in Algorithm 1, used as a subroutine in line 3. Compared to the variational inference, the $O(1)$ preferred model generation via convex combination in Algorithm 2 in line 7 is negligible. Therefore, the overall time complexity for $n$ tasks is $O(nv)$, regardless of preferred model generation. Moreover, as the memory complexity at each task is contributed by $O(1)$ memorization of posteriors by Algorithm 1, the total memory complexity is $O(n)$. Some of these posteriors will be discarded, as discussed in Section 4.1. Therefore, in the amortized case, Algorithm 3 ensures **sublinear buffer growth**.

The following theorem ensures that IBCL locates the user-preferred Pareto-optimal model with high probability.

**Theorem 2** (Probabilistic Pareto-optimality). *Pick any $\alpha \in [0, 1]$. The Pareto-optimal parameter $\theta_{\bar{w}}^{\star}$, i.e., the ground-truth parameter for $p_{\bar{w}}$, belongs to $\Theta_{\bar{w}}^{\alpha}$ with probability at least $1 - \alpha$ under distribution $\hat{q}_{\bar{w}}$. In formulas, $\mathrm{Pr}_{\theta_{\bar{w}}^{\star} \sim \hat{q}_{\bar{w}}} [\theta_{\bar{w}}^{\star} \in \Theta_{\bar{w}}^{\alpha}] \geq 1 - \alpha$.*

Theorem 2 gives us a $(1 - \alpha)$-guarantee in obtaining Pareto-optimal models for given task trade-off preferences. In other words, the Pareto-optimal parameter $\theta_{\bar{w}}^{\star}$ is guaranteed to belong to the Highest Density Region $\Theta_{\bar{w}}^{\alpha}$ that we build, with high probability. Our algorithm does not find the parameter $\theta_{\bar{w}}^{\star}$ itself, but instead the narrowest region $\Theta_{\bar{w}}^{\alpha}$ that contains it with high probability. In spirit, this result is very similar to what conformal prediction does (for predicted outputs, rather than parameters of interest) (Angelopoulos and Bates, 2022). Consequently, the IBCL algorithm enjoys the **probabilistic Pareto-optimality** targeted by our main problem. Please refer to Appendix G for the proof.

## 5 EXPERIMENTS

### 5.1 SETUP

Due to page limit, detailed setup can be found in Appendix H.

**Baselines.** *Although there exist many baseline methods for CL, only a few baselines for CLuST exist.* The following CLuST baselines are selected for comparison.

1. **Rehearsal-based.** This is the state-of-the-art technique for CLuST (Lin et al., 2019). These methods memorize a subset of training data of every task encountered so far. Task preferences are then given as weights to regularize the loss on each task's memorized data. We choose GEM (Lopez-Paz and Ranzato, 2017) and A-GEM (Chaudhry et al., 2018) as two typical rehearsal-based methods.
2. **Rehearsal-based, Bayesian.** Since IBCL is a Bayesian method, we also compare it to a Bayesian technique, VCL (Nguyen et al., 2018). We equip VCL with episodic memory to make it rehearsal-based, and so to be able to specify a preference. This approach has been used in (Servia-Rodriguez et al., 2021).
3. **Prompt-based.** *Prompt-based CL has never been used for CLuST and therefore not a state-of-the-art.* Still, they are considered efficient modern CL techniques. Therefore, we made an attempt to specify preferences on L2P (Wang et al., 2022), a prompt-based method, by training one learnable prompt prefix per task, and use a preference-weighted sum of the prompts at inference time.

**Benchmarks.** We experiment on four standard continual learning benchmarks, including three image classification and one NLP: (i) 15 tasks in CelebA (Liu et al., 2015) (with vs. without attributes), (ii) 10 tasks in Split CIFAR-100 (Zenke et al., 2017) (animals vs. non-animals), (iii) 10 tasks in TinyImageNet (Le and Yang, 2015) (animals vs. non-animals) and (iv) 5 tasks in 20NewsGroup (Lang, 1995) (news related to computers vs. not related to computers). For the first three image benchmarks, features are first extracted by ResNet-18 (He et al., 2016). For 20NewsGroup, features are extracted by TF-IDF (Aizawa, 2003). For each benchmark, all tasks share the same input and label space. There is no task id at training nor inference time, so the algorithm does not know from which task each data point comes from. Therefore, all experiments are set up as domain-incremental according to Van de Ven and Tolias (2019).

**Metrics.** As in standard continual learning evaluation, after training on task $i$, we evaluate the accuracy on all testing data of previous tasks $j \in \{1, \ldots, i\}$. To evaluate how well does a model address preferences, we randomly generate $n_{\mathrm{prefs}} = 10$ preferences per task, except for task 1, whose preference is always given by scalar 1. Therefore, for each method, we obtain 10 models at each task, and we evaluate a preference-weighted sum of their accuracies on previous tasks. Finally, these preference-weighted accuracies are used to compute standard continual learning metrics. The learning performance is measured by average per task accuracy and peak per task accuracy, and forgetting is measured by backward transfer (Díaz-Rodríguez et al., 2018).

**System.** Experiments are run on Intel(R) Core(TM) i7-8550U CPU @ 1.80GHz.

## 5.2 RESULTS

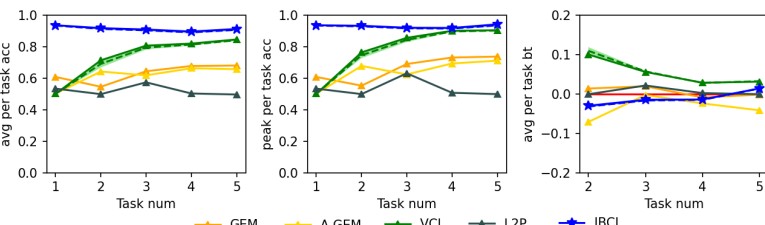

Figure 3: Results of 20NewsGroup. Since VCL and IBCL produce probabilistic models, we sample 10 deterministic models for each. The solid blue curve illustrates the top performance of deterministic models by IBCL, and the dashed blue curve is the mean performance. The shaded blue region is the performance range by IBCL. The same illustration method is used for VCL in green color.

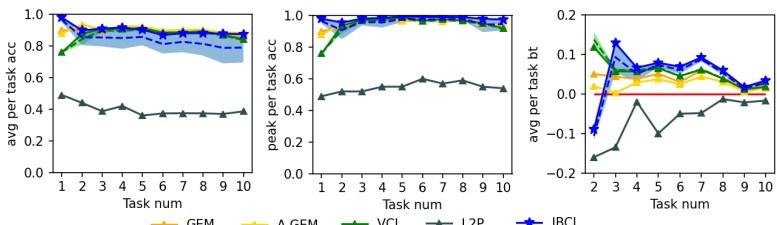

Figure 4: Results of TinyImageNet. The illustration method is the same as in Figure 3.

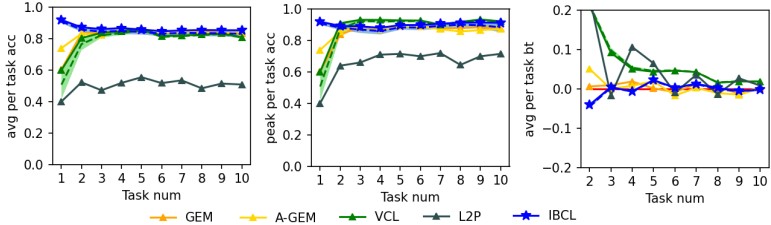

Figure 5: Results of Split CIFAR-100. The illustration method is the same as in Figure 3.

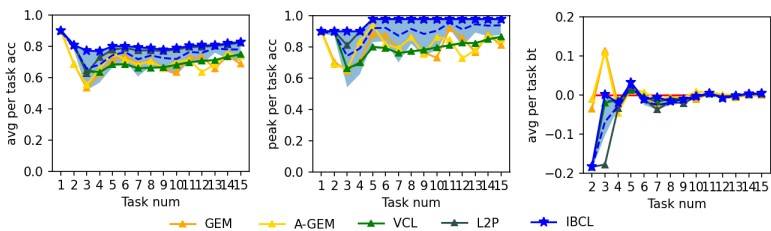

Figure 6: Results of CelebA. The illustration method is the same as in Figure 3.

**Main Results.** Our results support the claim that IBCL not only achieves high performance by probabilistic Pareto-optimality, but is also efficient with zero-shot generation of models.

Since VCL and IBCL output probabilistic models (BNNs and HDRs), we sample 10 deterministic models from each and compute the range of their performance metrics, illustrated as shaded areas in Figures 3, 4, 5, 6. They represent performances on the 20NewsGroup, TinyImageNet, Split CIFAR-100, and CelebA, respectively. In these figures, we draw the curves of top performance and mean performance of the sampled deterministic models by VCL and IBCL as solid and dashed lines, respectively.

Table 1: Training overhead comparison, measured as # of batch updates required at a task. Here, $n_i$: # of training data points at task $i$, $n_{\text{prefs}}$: # of preferences per task, $n_{\text{mem}}$: # of data points memorized per task in rehearsal, $n_{\text{priors}}$: # of priors in IBCL, $e$: # of epochs and $b$: batch size. Notice that the overhead of rehearsal based methods are proportional to $n_{\text{prefs}}$.

| | | # batch updates at task $i$ | # batch updates at last task | | | |
|---|---|---|---|---|---|---|
| | | | CelebA | CIFAR100 | TImgNet | 20News |
| Rehearsal | GEM A-GEM VCL | $n_{\text{prefs}}$ $\times (n_i + (i-1) \times n_{\text{mem}})$ $\times e/b$ | 99747 | 19532 | 13594 | 35313 |
| Prompt | L2P | $n_i \times e/b$ | 9538 | 1250 | 938 | 2907 |
| IBCL (ours) | | $n_{\text{priors}} \times n_i \times e/b$ | 28614 | 3750 | 2814 | 8721 |

From Figures 3, 4, 5 and 6, we can see that IBCL overall generates the model with top performance (high accuracy) in all cases, while maintaining little catastrophic forgetting (near-zero to positive backward transfer). This is due to the probabilistic Pareto-optimality guarantee. Statistically, IBCL improves on the baselines by at most 45% on average per task accuracy and by 43% on peak per task accuracy (compared to L2P in 20News). So far, to our knowledge, there is no discussion on how to specify a task trade-off preference in prompt-based continual learning, and we only make an attempt for L2P, which generally works poorly.

As illustrated in the figures, IBCL has a slightly negative backward transfer in the very beginning but then this value stays near-zero or positive. This shows that although IBCL may slightly forget the knowledge learned from the first task at the second task, it steadily retains knowledge afterwards. We can also see how, although VCL's backward transfer is higher than IBCL's in the first few tasks, it eventually decreases and takes values that are nearly identical to, or smaller than, the IBCL ones. For 20NewsGroup, this happens after 5 tasks, for TinyImageNet after 3 taks, for Split CIFAR-100 after 10 tasks, and for CelebA after 2 tasks.

Table 1 shows the training overhead comparison measured in number of batch updates per task. We can see how IBCL's overhead is independent of the number of preferences $n_{\text{prefs}}$ because it only requires training for the FGCS but not for the preferred models. **Consequently, our experiments show that IBCL is able to maintain a constant training overhead per task, regardless of $n_{\text{prefs}}$ while achieving high performance. Although L2P also has this constant overhead, its performance is too poor to be acceptable.**

**Ablation Studies.** The main experiments are conducted with hyperparameters $d = 0.002$, $\alpha = 0.01$, equal $\beta$'s, and prior distributions specified in Appendix H. We conduct ablation studies on $d$, $\alpha$, $\beta$ and priors. Due to page limit, details are in Appendix I.

## 6 CONCLUSION

Overall, we propose a probabilistic continual learning algorithm, namely IBCL, to tackle the CLuST problem, where an unbounded number of stability-plasticity trade-off preferences may be requested at each task. The design of IBCL enables not only learning performance, but also efficiency when solving the CLuST problem, as state-of-the-art methods require optimizing a loss function per preference, while IBCL only needs convex combinations. Such a benefit applies to various scales of models. It will be an interesting future direction to find a use case on large-scale models.

**Limitations of IBCL.** Poorly performing models can also be sampled from IBCL's HDRs. However, in practice, we can fine-tune $\alpha$ to shrink down the HDR to avoid poorly performing ones, as shown in the ablation studies. In addition, for a user-specified preference vector $\bar{w}$, probability $p_{\bar{w}}$ is the optimal distribution under entropy loss. One future research direction is to derive the preference vector $\bar{w}$ from some inputs. For example, we may learn this vector from an additional sequence of prompts (Wu et al., 2024). In that case, the preference vector itself might be different according to the design, including which loss is used.

**Broader Impacts.** IBCL is potentially useful in deriving user-customized models from large multi-task models, due to its not only guaranteed, but also zero-shot, preference addressing. These include large language models, recommendation systems and other applications.

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

## APPENDIX A    REASON TO ADOPT A BAYESIAN CONTINUAL LEARNING APPROACH

Let $q_0(\theta)$ be our prior probability density/mass function (pdf/pmf) on parameter $\theta \in \Theta$ at time $t = 0$. At time $t = 1$, we collect data $(\bar{x}_1, \bar{y}_1)$ pertaining to task 1, we elicit likelihood pdf/pmf $\ell_1(\bar{x}_1, \bar{y}_1 \mid \theta)$, and we compute $q_1(\theta \mid \bar{x}_1, \bar{y}_1) \propto q_0(\theta) \times \ell_1(\bar{x}_1, \bar{y}_1 \mid \theta)$. At time $t = 2$, we collect data $(\bar{x}_2, \bar{y}_2)$ pertaining to task 2 and we elicit likelihood pdf/pmf $\ell_2(\bar{x}_2, \bar{y}_2 \mid \theta)$. Now we have two options.

(i) Bayesian Continual Learning (BCL): we let the prior pdf/pmf at time $t = 2$ be the posterior pdf/pmf at time $t = 1$. That is, our prior pdf/pmf is $q_1(\theta \mid \bar{x}_1, \bar{y}_1)$, and we compute $q_2(\theta \mid \bar{x}_1, \bar{y}_1, \bar{x}_2, \bar{y}_2) \propto q_1(\theta \mid \bar{x}_1, \bar{y}_1) \times \ell_2(\bar{x}_2, \bar{y}_2 \mid \theta) \propto q_0(\theta) \times \ell_1(\bar{x}_1, \bar{y}_1 \mid \theta) \times \ell_2(\bar{x}_2, \bar{y}_2 \mid \theta)$;[1]

(ii) Bayesian Isolated Learning (BIL): we let the prior pdf/pmf at time $t = 2$ be a generic prior pdf/pmf $q'_0(\theta)$. We compute $q'_2(\theta \mid \bar{x}_2, \bar{y}_2) \propto q'_0(\theta) \times \ell_2(\bar{x}_2, \bar{y}_2 \mid \theta)$. We can even re-use the original prior, so that $q'_0 = q_0$.

---

[1]Here we tacitly assume that the likelihoods are independent.

As we can see, in option (i) we assume that the data generating process at time $t = 2$ takes into account both tasks, while in option (ii) we posit that it only takes into account task 2. Denote by $\sigma(X)$ the sigma-algebra generated by a generic random variable $X$. Let also $Q_2$ be the probability measure whose pdf/pmf is $q_2$, and $Q'_2$ be the probability measure whose pdf/pmf is $q'_2$. Then, we have the following.

**Proposition 1.** Posterior probability measure $Q_2$ can be written as a $\sigma(\bar{X}_1, \bar{Y}_1, \bar{X}_2, \bar{Y}_2)$-measurable random variable taking values in $[0, 1]$, while posterior probability measure $Q'_2$ can be written as a $\sigma(\bar{X}_2, \bar{Y}_2)$-measurable random variable taking values in $[0, 1]$.

*Proof.* Pick any $A \subset \Theta$. Then, $Q_2[A \mid \sigma(\bar{X}_1, \bar{Y}_1, \bar{X}_2, \bar{Y}_2)] = \mathbb{E}_{Q_2}[\mathbb{1}_A \mid \sigma(\bar{X}_1, \bar{Y}_1, \bar{X}_2, \bar{Y}_2)]$, a $\sigma(\bar{X}_1, \bar{Y}_1, \bar{X}_2, \bar{Y}_2)$-measurable random variable taking values in $[0, 1]$. Notice that $\mathbb{1}_A$ denotes the indicator function for set $A$. Similarly, $Q'_2[A \mid \sigma(\bar{X}_2, \bar{Y}_2)] = \mathbb{E}_{Q'_2}[\mathbb{1}_A \mid \sigma(\bar{X}_2, \bar{Y}_2)]$, a $\sigma(\bar{X}_2, \bar{Y}_2)$-measurable random variable taking values in $[0, 1]$. This is a well-known result in measure theory (Billingsley, 1986). $\square$

Of course Proposition 1 holds for all $t \geq 2$. Recall that the sigma-algebra $\sigma(X)$ generated by a generic random variable $X$ captures the idea of information encoded in observing $X$. An immediate corollary is the following.

**Corollary 1.** Let $t \geq 2$. Then, if we opt for BIL, we lose all the information encoded in $\{(\bar{X}_i, \bar{Y}_i)\}_{i=1}^{t-1}$.

In turn, if we opt for BIL, we obtain a posterior that is not measurable with respect to $\sigma(\{(\bar{X}_i, \bar{Y}_i)\}_{i=1}^{t}) \setminus \sigma(\bar{X}_t, \bar{Y}_t)$. If the true data generating process $p_t$ is a function of the previous data generating processes $p_{t'}$, $t' \leq t$, this leaves us with a worse approximation of the "true" posterior $Q^{\text{true}} \propto Q_0 \times p_t$.

The phenomenon in Corollary 1 is commonly referred to as *catastrophic forgetting*. Continual learning literature is unanimous in labeling catastrophic forgetting as undesirable – see e.g. (Farquhar and Gal, 2019; Li et al., 2020). For this reason, in this work we adopt a BCL approach. In practice, we cannot compute the posterior pdf/pmf exactly, and we will resort to variational inference to approximate them – an approach often referred to as Variational Continual Learning (VCL) (Nguyen et al., 2018). As we shall see in Appendix E, Assumption 1 is needed in VCL to avoid catastrophic forgetting.

## A.1 Relationship between IBCL and other BCL techniques

Like (Farquhar and Gal, 2019; Li et al., 2020), the weights in our Bayesian neural networks (BNNs) have Gaussian distribution with diagonal covariance matrix. Because IBCL is rooted in Bayesian continual learning, we can initialize IBCL with a much smaller number of parameters to solve a complex task as long as it can solve a set of simpler tasks. In addition, IBCL does not need to evaluate the importance of parameters by measures such as computing the Fisher information, which are computationally expensive and intractable in large models.

### A.1.1 Relationship between IBCL and MAML

In this section, we discuss the relationship between IBCL and the Model-Agnostic Meta-Learning (MAML) and Bayesian MAML (BMAML) procedures introduced in (Finn et al., 2017; Yoon et al., 2018), respectively. These are inherently different than IBCL, since the latter is a continual learning procedure, while MAML and BMAML are meta-learning algorithms. Nevertheless, given the popularity of these procedures, we feel that relating IBCL to them would be useful to draw some insights on IBCL itself.

In MAML and BMAML, a task $i$ is specified by a $n_i$-shot dataset $D_i$ that consists of a small number of training examples, e.g. observations $(x_{1_i}, y_{1_i}), \ldots, (x_{n_i}, y_{n_i})$. Tasks are sampled from a task distribution $\mathbb{T}$ such that the sampled tasks share the statistical regularity of the task distribution. In IBCL, Assumption 1 guarantees that the tasks $p_i$ share the statistical regularity of class $\mathcal{F}$. MAML and BMAML leverage this regularity to improve the learning efficiency of subsequent tasks.

At each meta-iteration $i$,

1. *Task-Sampling*: For both MAML and BMAML, a mini-batch $T_i$ of tasks is sampled from the task distribution $\mathbb{T}$. Each task $\tau_i \in T_i$ provides task-train and task-validation data, $D_{\tau_i}^{\text{trn}}$ and $D_{\tau_i}^{\text{val}}$, respectively.

2. *Inner-Update*: For MAML, the parameter of each task $\tau_i \in T_i$ is updated starting from the current generic initial parameter $\theta_0$, and then performing $n_i$ gradient descent steps on the task-train loss. For BMAML, the posterior $q(\theta_{\tau_i} \mid D_{\tau_i}^{\text{trn}}, \theta_0)$ is computed, for all $\tau_i \in T_i$.

3. *Outer-Update*: For MAML, the generic initial parameter $\theta_0$ is updated by gradient descent. For BMAML, it is updated using the Chaser loss (Yoon et al., 2018, Equation (7)).

Notice how in our work $\bar{w}$ is a probability vector. This implies that if we fix a number of task $k$ and we let $\bar{w}$ be equal to $(w_1, \ldots, w_k)^\top$, then $\bar{w} \cdot \bar{p}$ can be seen as a sample from $\mathbb{T}$ such that $\mathbb{T}(p_i) = w_i$, for all $i \in \{1, \ldots, k\}$.

Here lies the main difference between IBCL and BMAML. In the latter the information provided by the tasks is used to obtain a refinement of the (parameter of the) distribution $\mathbb{T}$ on the tasks themselves. In IBCL, instead, we are interested in the optimal parameterization of the posterior distribution associated with $\bar{w} \cdot \bar{p}$. Notice also that at time $k+1$, in IBCL the support of $\mathbb{T}$ changes: it is $\{p_1, \ldots, p_{k+1}\}$, while for MAML and BMAML it stays the same.

Also, MAML and BMAML can be seen as ensemble methods, since they use different values (MAML) or different distributions (BMAML) to perform the Outer-Update and come up with a single value (MAML) or a single distributions (BMAML). Instead, IBCL keeps distributions separate via FGCS, thus capturing the ambiguity faced by the designer during the analysis.

Furthermore, we want to point out how while for BMAML the tasks $\tau_i$ are all "candidates" for the true data generating process (dgp) $p_i$, in IBCL we approximate the pdf/pmf of $p_i$ with the product $\prod_{h=1}^{i} \ell_h$ of the likelihoods up to task $i$. The idea of different candidates for the true dgp is beneficial for IBCL as well: in the future, we plan to let go of Assumption 1 and let each $p_i$ belong to a credal set $\mathcal{P}_i$. This would capture the epistemic uncertainty faced by the agent on the true dgp.

To summarize, IBCL is a continual learning technique whose aim is to find the correct parameterization of the posterior associated with $\bar{w} \cdot \bar{p}$. Here, $\bar{w}$ expresses the developer's preferences on the tasks. MAML and BMAML, instead, are meta-learning algorithms whose main concern is to refine the distribution $\mathbb{T}$ from which the tasks are sampled. While IBCL is able to capture the preferences of, and the ambiguity faced by, the designer, MAML and BMAML are unable to do so. On the contrary, these latter seem better suited to solve meta-learning problems. An interesting future research direction is to come up with imprecise BMAML, or IBMAML, where a credal set $\text{Conv}(\{\mathbb{T}_1, \ldots, \mathbb{T}_k\})$ is used to capture the ambiguity faced by the developer in specifying the correct distribution on the possible tasks. The process of selecting one element from such credal set may lead to computational gains.

## APPENDIX B    HIGHEST DENSITY REGION

Equivalently to Definition 2, an HDR is defined as follows (Hyndman, 1996).

**Definition 5.** Let $\Theta$ be a set of interest, and consider a significance level $\alpha \in [0, 1]$. Suppose that a (continuous) random variable $\theta \in \Theta$ has probability density function (pdf) $q$.[2] The *$\alpha$-level Highest Density Region (HDR)* $\Theta_q^\alpha$ is the subset of $\Theta$ such that

$$\Theta_q^\alpha = \{\theta \in \Theta : q(\theta) \geq q^\alpha\}, \tag{4}$$

where $q^\alpha$ is a constant value. In particular, $q^\alpha$ is the largest constant such that $\Pr_{\theta \sim q}[\theta \in \Theta_q^\alpha] \geq 1 - \alpha$.

Some scholars indicate HDRs as the Bayesian counterpart to the frequentist concept of confidence intervals. In dimension 1, $\Theta_q^\alpha$ can be interpreted as the narrowest interval – or union of intervals – in which the value of the (true) parameter falls with probability of at least $1 - \alpha$ according to distribution $q$. We give a simple visual example in Figure 7.

---

[2]Here too, for ease of notation, we do not distinguish between a random variable and its realization.

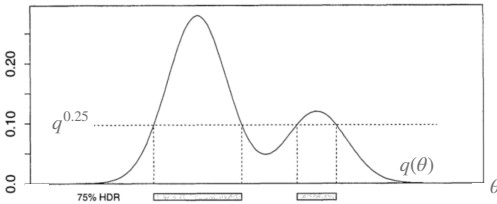

Figure 7: The 0.25-HDR for a Normal Mixture density. This picture is a replica of Figure 1 in (Hyndman, 1996). The geometric representation of "75% probability according to $q$" is the area between the pdf curve $q(\theta)$ and the horizontal bar corresponding to $q^{0.25}$. A higher probability coverage (according to $q$) would correspond to a lower constant, so $q^{\alpha} < q^{0.25}$, for all $\alpha < 0.25$. In the limit, we recover 100% coverage at $q^0 = 0$.

## APPENDIX C  ADDITIONAL RELATED WORK

**Multi-task Learning under Preferences.** Learning for Pareto-optimal models under task performance trade-offs has been studied by researchers in multi-task learning (Caruana, 1997; Sener and Koltun, 2018). Various techniques have been applied to obtain models that address particular trade-off points (Lin et al., 2019; 2020; Ma et al., 2020; Gupta et al., 2021). The idea of preferences on the trade-off points is introduced in multi-objective optimization (Lin et al., 2020; Sener and Koltun, 2018), and a preference can guide learning algorithms to search for a particular model. We borrow the formalization of preferences from (Mahapatra and Rajan, 2020), where a preference is given by a vector of non-negative real weights $\bar{w}$, with each entry $w_i$ corresponding to task $i$. That is, $w_i \geq w_j \iff i \succeq j$. This means that if $w_i \geq w_j$, then task $i$ is preferred to task $j$. However, state-of-the-art algorithms require training one model per preference, imposing large overhead when there is a large number of preferences.

**Continual Learning.** Continual learning, also known as lifelong learning, is a special case of multi-task learning, where tasks arrive sequentially instead of simultaneously (Thrun, 1998; Ruvolo and Eaton, 2013a). In this paper, we leverage Bayesian inference in the knowledge base update (Ebrahimi et al., 2019). Like generic multi-task learning, continual learning also faces the stability-plasticity trade-off (De Lange et al., 2021), which balances between performance on new tasks and resistance to catastrophic forgetting (Kirkpatrick et al., 2017). Current methods identify models to address trade-off preferences by techniques such as loss regularization (Servia-Rodriguez et al., 2021), meaning at least one model needs to be trained per preference.

Researchers in CL have proposed various approaches to retain knowledge while updating a model on new tasks. These include modified loss landscapes for optimization (Farajtabar et al., 2020), preservation of critical pathways via attention (Abati et al., 2020), memory-based methods (Lopez-Paz and Ranzato, 2017), shared representations (Lee et al., 2019), and dynamic representations (Bulat et al., 2020). Bayesian, or probabilistic methods such as variational inference are also adopted (Ebrahimi et al., 2019; Nguyen et al., 2018).

In BCL (Nguyen et al., 2018), each task is associated with a data generating process (likelihood, elicited according to the data at hand), parameterized by $\theta$. The latter is postulated to be a random quantity, which at the beginning of the analysis has a prior distribution, $\theta \sim q_0$. After training on the available data, the prior distribution is turned into posterior, $\theta \sim q_1$ via Bayes' theorem. The posterior $q_1$ is the revised parameter distribution after having learned from the data pertaining to the first task to complete. It is then used as a prior for the next task.

## APPENDIX D  2-WASSERSTEIN METRIC

In the main portion of the paper, we endowed $\Delta_{\mathcal{X}\mathcal{Y}}$ with the 2-Wasserstein metric. It is defined as

$$\|p - p'\|_{W_2} \equiv W_2(p, p') := \sqrt{\inf_{\gamma \in \Gamma(p,p')} \mathbb{E}_{((x_1,y_1),(x_2,y_2)) \sim \gamma}[d((x_1, y_1), (x_2, y_2))^2]}, \quad (5)$$

where

1. $p, p' \in \Delta_{\mathcal{X}\mathcal{Y}}$;

2. $\Gamma(p, p')$ is the set of all couplings of $p$ and $p'$. A coupling $\gamma$ is a joint probability measure on $(\mathcal{X} \times \mathcal{Y}) \times (\mathcal{X} \times \mathcal{Y})$ whose marginals are $p$ and $p'$ on the first and second factors, respectively;

3. $d$ is the product metric endowed to $\mathcal{X} \times \mathcal{Y}$ (Deza and Deza, 2013, Section 4.2).[3]

We choose the 2-Wasserstein distance for the ease of computation. In practice, when all distributions are modeled by Bayesian neural networks with independent Gaussian weights and biases, we have

$$\|q_1 - q_2\|_{W_2}^2 = \|\mu_{q_1}^2 - \mu_{q_2}^2\|_2^2 + \|\sigma_{q_1}^2 \mathbf{1} - \sigma_{q_2}^2 \mathbf{1}\|_2^2, \tag{6}$$

where $\| \cdot \|_2$ denotes the Euclidean norm, $\mathbf{1}$ is a vector of all 1's, and $\mu_q$ and $\sigma_q$ are respectively the mean and standard deviation of a multivariate normal distribution $q$ with independent dimensions, $q = \mathcal{N}(\mu_q, \sigma_q^2 I)$, $I$ being the identity matrix. Therefore, computing the $W_2$-distance between two distributions is equivalent to computing the difference between their means and variances, which costs O(1) in time complexity.

## APPENDIX E    IMPORTANCE OF ASSUMPTION 1

We need Assumption 1 in light of the results in (Kessler et al., 2023). There, the authors show that misspecified models can forget even when Bayesian inference is carried out exactly. By requiring that $\mathrm{diam}(\mathcal{F}) = r$, we control the amount of misspecification via $r$. In (Kessler et al., 2023), the authors design a new approach – called Prototypical Bayesian Continual Learning, or ProtoCL – that allows dropping Assumption 1 while retaining the Bayesian benefit of remembering previous tasks. Because the main goal of this paper is to come up with a procedure that allows the designer to express preferences over the tasks, we retain Assumption 1, and we work in the classical framework of Bayesian Continual Learning. In the future, we plan to generalize our results by operating with ProtoCL.[4]

## APPENDIX F    AN EXAMPLE OF A PARAMETERIZED FAMILY $\mathcal{F}$

Let us give an example of a parameterized family $\mathcal{F}$. Suppose that we have one-dimensional data points and labels. At each task $i$, the marginal on $\mathcal{X}$ of $p_i$ is a Normal $\mathcal{N}(\mu, 1)$, while the conditional distribution of label $y \in \mathcal{Y}$ given data point $x \in \mathcal{X}$ is a categorical $\mathrm{Cat}(\vartheta)$. Hence, the parameter for $p_i$ is $\theta = (\mu, \vartheta)$, and it belongs to $\Theta = \mathbb{R} \times \mathbb{R}^{|\mathcal{Y}|}$. In this situation, an example of a family $\mathcal{F}$ satisfying Assumptions 1 and 2 is the convex hull of distributions that can be decomposed as we just described, and whose distance according to the 2-Wasserstein metric does not exceed some $r > 0$.

## APPENDIX G    PROOFS OF THE THEOREMS

*Proof of Theorem 1.* Without loss of generality, suppose we have encountered $i = 2$ tasks so far, so the FGCS is $\mathcal{Q}_2$. Let $\mathrm{ex}[\mathcal{Q}_1] = \{q_1^j\}_{j=1}^{m_1}$ and $\mathrm{ex}[\mathcal{Q}_2] \setminus \mathrm{ex}[\mathcal{Q}_1] = \{q_2^j\}_{j=1}^{m_2}$. Let $\hat{q}$ be any element of $\mathcal{Q}_2$. Then, there exists a probability vector $\bar{\beta} = (\beta_1^1, \ldots, \beta_1^{m_1}, \beta_2^1, \ldots, \beta_2^{m_2})^\top$ such that

$$\hat{q} = \sum_{j=1}^{m_1} \beta_1^j q_1^j + \sum_{j=1}^{m_2} \beta_2^j q_2^j \propto \hat{p}_1 \sum_{j=1}^{m_1} \beta_1^j q_0^j + \hat{p}_2 \sum_{j=1}^{m_2} \beta_2^j q_0^j. \tag{7}$$

Here, $\hat{p}_i = \prod_{k=1}^{i} \ell_k$, and $\ell_k$ is the likelihood at task $k$. It estimates the pdf of the true data generating process $p_i$ of task $i$. The proportional relationship in equation 7 is based on the Bayesian inference step (line 3, approximated via variational inference) of Algorithm 1. We can then find a vector $\bar{w} = (w_1 = \sum_{j=1}^{m_1} \beta_1^j, w_2 = \sum_{j=1}^{m_2} \beta_2^j)^\top$ that expresses the designer's preferences over tasks 1 and 2. As we can see, then, the act of selecting a generic distribution $\hat{q} \in \mathcal{Q}_2$ is equivalent to specifying a preference vector $\bar{w}$ over tasks 1 and 2. This concludes the proof. $\qquad\square$

---

[3]We denote by $d_\mathcal{X}$ and $d_\mathcal{Y}$ the metrics endowed to $\mathcal{X}$ and $\mathcal{Y}$, respectively.

[4]In (Kessler et al., 2023), the authors also show that if there is a task dataset imbalance, then the model can forget under certain assumptions. To avoid complications, in this work we tacitly assume that task datasets are balanced.

*Proof of Theorem 2.* For maximum generality, assume $\Theta$ is uncountable. Recall from Definition 2 that $\alpha$-*level Highest Density Region* $\Theta_{\bar{w}}^{\alpha}$ is defined as the subset of the parameter space $\Theta$ such that

$$\int_{\Theta_{\bar{w}}^{\alpha}} \hat{q}_{\bar{w}}(\theta)\mathrm{d}\theta \geq 1 - \alpha \quad \text{and} \quad \int_{\Theta_{\bar{w}}^{\alpha}} \mathrm{d}\theta \text{ is a minimum.}$$

We need $\int_{\Theta_{\bar{w}}^{\alpha}} \mathrm{d}\theta$ to be a minimum because we want $\Theta_{\bar{w}}^{\alpha}$ to be the smallest possible region that gives us the desired probabilistic coverage. Equivalently, from Definition 5 we can write that $\Theta_{\bar{w}}^{\alpha} = \{\theta \in \Theta : \hat{q}_{\bar{w}}(\theta) \geq \hat{q}_{\bar{w}}^{\alpha}\}$, where $\hat{q}_{\bar{w}}^{\alpha}$ is the largest constant such that $\Pr_{\theta \sim \hat{q}_{\bar{w}}}[\theta \in \Theta_{\bar{w}}^{\alpha}] \geq 1 - \alpha$. Our result $\Pr_{\theta_{\bar{w}}^{\star} \sim \hat{q}_{\bar{w}}}[\theta_{\bar{w}}^{\star} \in \Theta_{\bar{w}}^{\alpha})] \geq 1 - \alpha$, then, comes from the fact that $\Pr_{\theta_{\bar{w}}^{\star} \sim \hat{q}_{\bar{w}}}[\theta_{\bar{w}}^{\star} \in \Theta_{\bar{w}}^{\alpha})] = \int_{\Theta_{\bar{w}}^{\alpha}} \hat{q}_{\bar{w}}(\theta)\mathrm{d}\theta$, a consequence of a well-known equality in probability theory (Billingsley, 1986). $\square$

## APPENDIX H  DETAILS OF EXPERIMENTS

We select 15 tasks from CelebA. All tasks are binary image classification on celebrity face images. Each task $i$ is to classify whether the face has an attribute such as wearing eyeglasses or having a mustache. The first 15 attributes (out of 40) in the attribute list (Liu et al., 2015) are selected for our tasks. The training, validation and testing sets are already split upon download, with 162,770, 19,867 and 19,962 images, respectively. All images are annotated with binary labels of the 15 attributes in our tasks. We use the same training, validation and testing set for all tasks, with labels being the only difference.

We select 20 classes from CIFAR100 (Krizhevsky et al., 2009) to construct 10 Split-CIFAR100 tasks (Zenke et al., 2017). Each task is a binary image classification between an animal class (label 0) and a non-animal class (label 1). The classes are (in order of tasks):

1. Label 0: aquarium fish, beaver, dolphin, flatfish, otter, ray, seal, shark, trout, whale.
2. Label 1: bicycle, bus, lawn mower, motorcycle, pickup truck, rocket, streetcar, tank, tractor, train.

That is, the first task is to classify between aquarium fish images and bicycle images, and so on. We want to show that the continual learning model incrementally gains knowledge of how to identify animals from non-animals throughout the task sequence. For each class, CIFAR100 has 500 training data points and 100 testing data points. We hold out 100 training data points for validation. Therefore, at each task we have $400 \times 2 = 800$ training data, $100 \times 2 = 200$ validation data and $100 \times 2 = 200$ testing data.

We also select 20 classes from TinyImageNet (Le and Yang, 2015). The setup is similar to Split-CIFAR100, with label 0 being animals and 1 being non-animals.

1. Label 0: goldfish, European fire salamander, bullfrog, tailed frog, American alligator, boa constrictor, goose, koala, king penguin, albatross.
2. Label 1: cliff, espresso, potpie, pizza, meatloaf, banana, orange, water tower, via duct, tractor.

The dataset already splits 500, 50 and 50 images for training, validation and testing per class. Therefore, each task has 1000, 100 and 100 images for training, validation and testing, respectively.

20NewsGroups (Lang, 1995) contains news report texts on 20 topics. We select 10 topics for 5 binary text classification tasks. Each task is to distinguish whether the topic is computer-related (label 0) or not computer-related (label 1), as follows.

1. Label 0: comp.graphics, comp.os.ms-windows.misc, comp.sys.ibm.pc.hardware, comp.sys.mac.hardware, comp.windows.x.
2. Label 1: misc.forsale, rec.autos, rec.motorcycles, rec.sport.baseball, rec.sport.hockey.

Each class has different number of news reports. On average, a class has 565 reports for training and 376 for testing. We then hold out 100 reports from the 565 for validation. Therefore, each binary classification task has 930, 200 and 752 data points for training, validation and testing, on average respectively.

All data points are first preprocessed by a feature extractor. For images, the feature extractor is a pre-trained ResNet18 (He et al., 2016). We input the images into the ResNet18 model and obtain its last hidden layer's activations, which has a dimension of 512. For texts, the extractor is TF-IDF (Aizawa, 2003) succeeded with PCA to reduce the dimension to 512 as well.

Each Bayesian network model is trained with evidence lower bound (ELBO) loss, with a fixed feed-forward architecture (input=512, hidden=64, output=1). The hidden layer is ReLU-activated and the output layer is sigmoid-activated. Therefore, our parameter space $\Theta$ is the set of all values that can be taken by this network's weights and biases.

The three variational inference priors, learning rate, batch size and number of epcohs are tuned on validation sets. The tuning results are as follows. Here, "lr" stands for learning rate.

1. CelebA: priors = $\{\mathcal{N}(0, 0.2^2 I), \mathcal{N}(0, 0.25^2 I), \mathcal{N}(0, 0.3^2 I)\}$, lr = $1e-3$, batch size = 64, epochs = 10.
2. Split-CIFAR100: priors = $\{\mathcal{N}(0, 2^2 I), \mathcal{N}(0, 2.5^2 I), \mathcal{N}(0, 3^2 I)\}$, lr = $5e-4$, batch size = 32, epochs = 50.
3. TinyImageNet: priors = $\{\mathcal{N}(0, 2^2 I), \mathcal{N}(0, 2.5^2 I), \mathcal{N}(0, 3^2 I)\}$, lr = $5e-4$, batch size = 32, epochs = 30.
4. 20NewsGroup: priors = $\{\mathcal{N}(0, 2^2 I), \mathcal{N}(0, 2.5^2 I), \mathcal{N}(0, 3^2 I)\}$, lr = $5e-4$, batch size = 32, epochs = 100.

For the baseline methods, we use exactly the same learning rate, batch sizes and epochs. For probabilistic baseline methods (VCL), we use the prior with the median standard deviation. For example, on CelebA tasks, VCL uses the normal prior $\mathcal{N}(0, 0.25^2 I)$.

For rehearsal-based baselines, the memory size per task for CelebA is 200, and for the rest is 50. Together with the numbers above, we can compute the numerical values in Table 1.

## APPENDIX I    ABLATION STUDIES

### I.1    DIFFERENT $d$'S

Here, we evaluate the effects of choosing different thresholds $d$. We experiment on 20NewsGroup and Split-CIFAR100. The variations include

1. $d = 0.02$, same as in the main experiments.
2. $d = 0.05$.
3. $d = 0.08$.

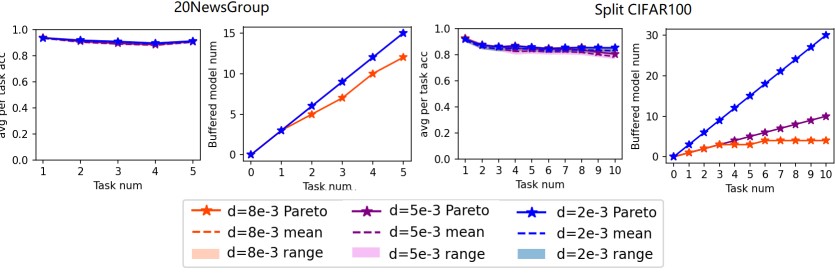

Figure 8: Different $d$'s on 20NewsGroup and Split-CIFAR100. The buffer growth curves of $d = 5e-3$ and $d = 2e-3$ of 20NewsGroup are overlapping.

As $d$ increases, we are allowing more posteriors in the knowledge base to be reused. This will lead to memory efficiency at the cost of a performance drop. Figure 8 supports this trend. With an appropriately selected $d$, we can guarantee that the model's performance will not be overly affected, and that we save buffer memory. For Split-CIFAR100, when $d = 8e-3$, the buffer stops growing after task 6.

## I.2 DIFFERENT $\alpha$'S

Here, we evaluate the effects of choosing different significance level $\alpha$. We experiment on 20News-Group and Split-CIFAR100. The variations include

1. $\alpha = 0.01$, same as the main experiments.
2. $\alpha = 0.1$.
3. $\alpha = 0.25$.

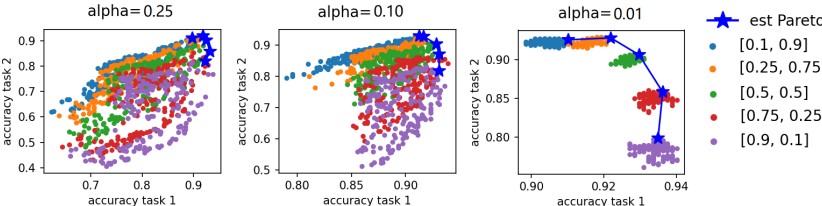

Figure 9: Different $\alpha$'s on different preferences over the first two tasks in 20NewsGroup.

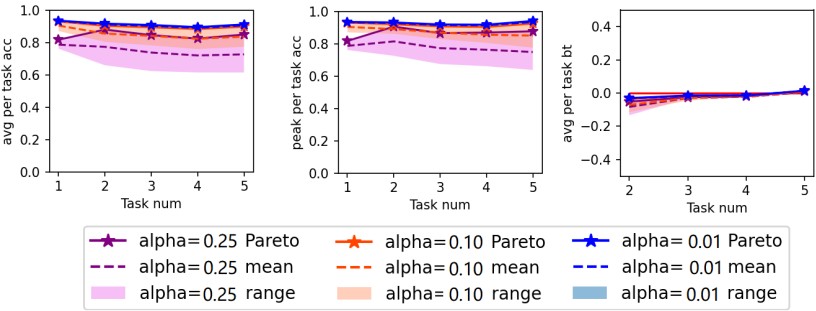

Figure 10: Different $\alpha$'s on randomly generated preferences over all tasks in 20NewsGroup.

In Figure 9, we evaluate testing accuracy on three different $\alpha$'s over five different preferences (from $[0.1, 0.9]$ to $[0.9, 0.1]$) on the first two tasks of 20NewsGroup. For each preference, we uniformly sample 200 deterministic models from the HDR. We use the sampled model with the maximum L2 sum of the two accuracies to estimate the Pareto optimality under a preference. We can see that, as $\alpha$ approaches 0, we tend to sample closer to the Pareto front. This is because, with a smaller $\alpha$, HDRs become wider and we have a higher probability to sample Pareto-optimal models according to Theorem 2. For instance, when $\alpha = 0.01$, we have a probability of at least 0.99 that the Pareto-optimal solution is contained in the HDR. Figure 10 shows that the performance drops as $\alpha$ increases, because we are more likely to sample poorly performing models from the HDR.

## I.3 DIFFERENT PRIORS

Here, we evaluate the effects of different priors. We experiment on 20NewsGroup and Split-CIFAR100. We first evaluate different sizes of prior standard deviations.

1. Medium prior stds = $\{2, 2.5, 3\}$, same as the main experiments.
2. Small prior stds = $\{0.2, 0.25, 0.3\}$.
3. Large prior stds = $\{20, 25, 30\}$.

Figure 11 and 12 show the effects of different prior std sizes on the learning performance, on 20NewsGroup and Split-CIFAR100, respectively. We can see that in the beginning of 20News-Group, small prior stds lower the average and peak per task accuracy. However, this decrease in performance is gradually reduced, and eliminated at tasks 4 and 5. This is because the posterior distributions gradually approach to a pdf that contains the Pareto-optimal model with high probability. The forgetting prevention of the small stds is slightly improved from the other two trials, with a

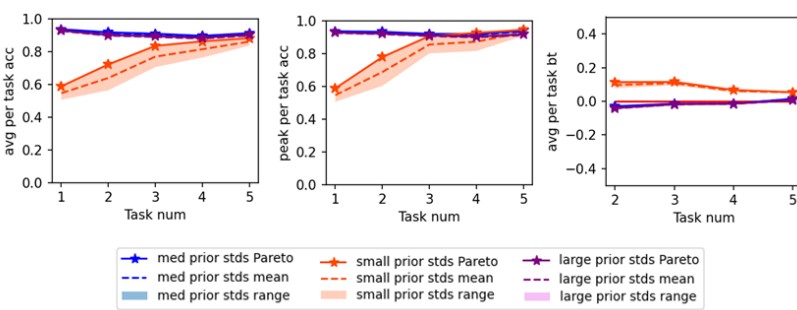

Figure 11: Different prior std sizes on randomly generated preferences over all tasks in 20NewsGroup.

In BCL, each task is associated with a data generating process (likelihood, elicited according to the data at hand), parameterized by $\theta$. The latter is postulated to be a random quantity, which at the beginning of the analysis has a prior distribution, $\theta \sim q_0$. After training on the available data, the prior distribution is turned into posterior, $\theta \sim q_1$ via Bayes' theorem. The posterior $q_1$ is the revised parameter distribution after having learned from the data pertaining to the first task to complete. It is then used as a prior for the next task.

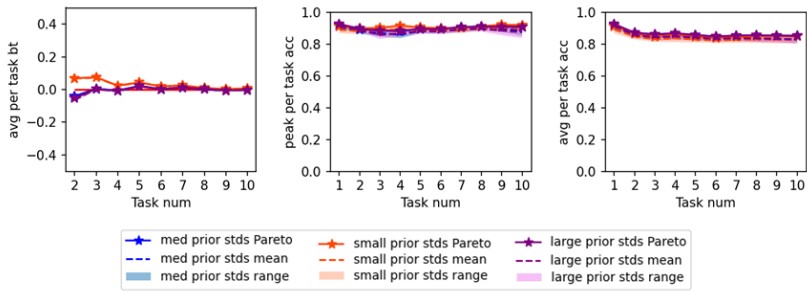

Figure 12: Different prior std sizes on randomly generated preferences over all tasks in Split-CIFAR100.

higher backward transfer initially, but also merges with the others at the end due to the same reason. All backward transfers are near zero, meaning there is almost no forgetting. For Split-CIFAR100, the same divergence in performance appear in the beginning 3 tasks, most obvious in average accuracy. Then, the same pattern follows.

We conclude that different choices of priors may lower the performance in the initial tasks, but the performance will gradually improve and align with each other to achieve Pareto-optimality. Next, we evaluate different numbers of priors.

1. 3 priors, stds = $\{2, 2.5, 3\}$, same as the main experiments.
2. 5 priors, stds = $\{1.5, 2, 2.5, 3, 3.5\}$.
3. 8 priors, stds = $\{1, 1.5, 2, 2.5, 3, 3.5, 4\}$.

As shown in Figure 13 and 14, different numbers of priors (3, 5 and 7) show very similar trends in performance. We therefore conclude that a small number of 3 priors is sufficient.

### I.4 DIFFERENT $\beta$'S

We also evaluate the effects of different $\beta$'s. On 20NewsGroup and Split-CIFAR100, we have

1. Equal $\beta$'s, same as the main experiments.
2. Randomized $\beta$'s.

As shown in Figure 15 and 16, equal sized $\beta$'s and randomly split sizes of $\beta$'s have almost the same performance trends. This provides evidence for our statement in Section 4.2, that using different

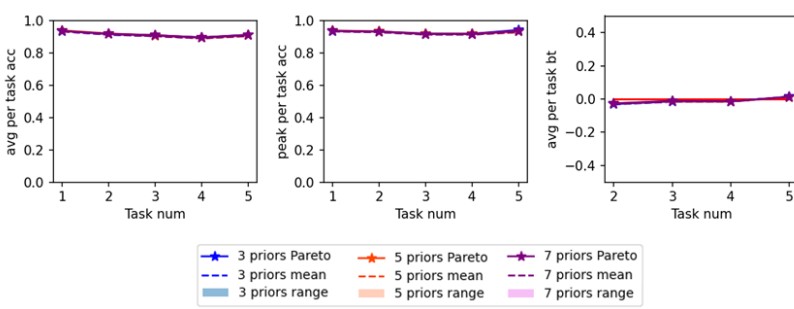

Figure 13: Different numbers of priors on randomly generated preferences over all tasks in 20NewsGroup.

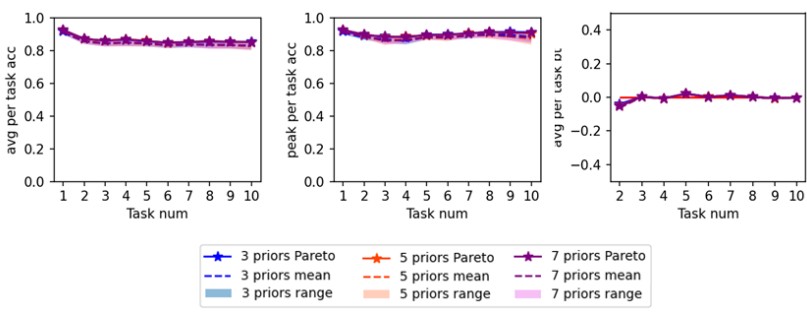

Figure 14: Different numbers of priors on randomly generated preferences over all tasks in Split-CIFAR100.

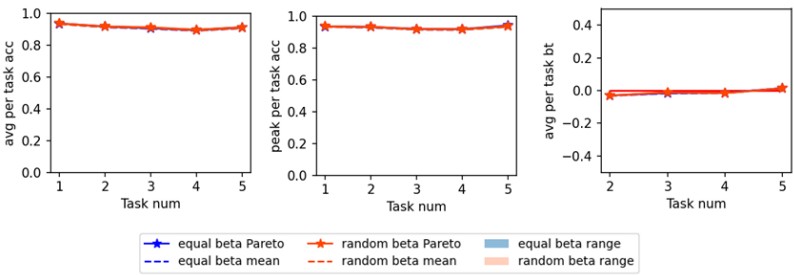

Figure 15: Equal and randomized $\beta$'s on randomly generated preferences over all tasks in 20NewsGroup.

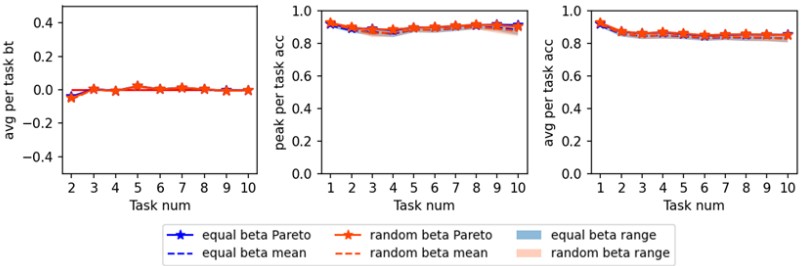

Figure 16: Equal and randomized $\beta$'s on randomly generated preferences over all tasks in Split-CIFAR100.

choices of $\beta$'s would not affect the overall performance. Therefore, using equal sized $\beta$'s would be sufficient.

