# OpenReview forum: "IBCL: Zero-shot Model Generation under Stability-Plasticity Trade-offs"
_ICLR.cc/2025/Conference — Submitted to ICLR 2025_

### Official Review · Reviewer_qVr1 · 2024-10-27

**Soundness:** 3
**Presentation:** 4
**Contribution:** 3
**Rating:** 6
**Confidence:** 3

**Summary:**

The paper introduces Imprecise Bayesian Continual Learning (IBCL), a zero-shot, Pareto-optimal model with sublinear buffer growth, designed to address the Continual Learning under Specific Trade-offs (CLuST) problem. It also provides a mathematical formulation of the CLuST problem and presents experiments to evaluate IBCL's effectiveness.

**Strengths:**

The paper is well-written and easy to follow. The mathematical formulation of the problem is clear, and the figures effectively illustrate the proposed algorithm. Additionally, the paper provides analytical insights into the proposed algorithm, and the experimental results demonstrate its effectiveness.

**Weaknesses:**

[1]I'm not sure about the applicability of the problem setup and the proposed model. It seems that the target distribution for learning (line 188) is only optimal under entropy loss for predictions. For more details, please refer to the questions section.

[2]A minor issue is the lack of ablation studies on the choices of prior distributions $q_0^j$'s and the parameter $\beta$'s. In the experiments, prior choices (lines 968-975) are fixed, and the $\beta$ values are uniformly chosen to recover the preference vector (Algorithm 2). Ablation studies on these choices would provide helpful insights.

**Questions:**

I’m curious about the “optimality” of the distribution to learn $p_{\bar{w}}$. Given a preference vector, the mixed probability $p_{\bar{w}}=\sum w_i p_i$ seems optimal under entropy loss; specifically, if $(X,Y)\sim p_{\bar{w}}$  then the probability $p_{\bar{w}}$ (conditional on $X$) minimizes the entropy loss when predicting $Y$ given $X$. However, if we consider an L2 loss, the optimal prediction becomes $\mathbb{E}[Y|X]$, a point estimate rather than a distribution. My questions are:

(1) Is the convex combination of true distributions $p_{\bar{w}}=\sum w_i p_i$ only optimal under entropy loss?

(2) If so, could the current method be extended to accommodate other loss functions, such as L2  or absolute loss?

Also, in the broader impact section, the potential of the proposed approach for large language models is mentioned. However, the experiments are conducted with a small neural network with only a single hidden layer. How does IBCL perform in terms of efficiency and effectiveness, particularly in training time per new task, when applied to larger models?

---

> ### Author Response · Authors · 2024-11-19
>
> We appreciate the reviewer for the positive rating. We address the reviewer’s concerns as follows. If any concerns still persist, we are more than happy to discuss them and edit our paper accordingly. We will be very grateful if the reviewer considers improving the rating.
>
> 1. It will be best to have additional ablation studies.
>
> We thank the reviewer for pointing this out. We have added additional ablation studies on different priors and different choices of $beta$ in Appendix I.
>
> 2. Does IBCL still have benefits when applying to larger models?
>
> When solving CLuST problems, IBCL does have benefits in efficiency compared to state-of-the-art methods. As current methods require running an optimization of a loss function when generating a model, while IBCL only needs a convex combination. This benefit applies to various scales of models. Still, it will be an interesting future research to identify a use case on large-scale models. This is mentioned in our updated Section 6.
>
> 3. For the optimality of $p_{\bar{w}}$:
> (1) Is the convex combination of true distributions only under cross entropy loss?
> (2) If so, can it be extended to other loss functions, such as L2 or absolute loss?
>
> We thank the reviewer for these insightful questions. It is indeed true that $p_{\bar{w}}$ minimizes the entropy loss. We did not think of the possibility of other losses because we treated the preference as given by/known to the user. One potential research direction is to generalize IBCL, so that it can derive the preference vector $\bar{w}$ from some inputs. For example, we may learn this preference from additional sequential prompts [1]. In that case, the preference vector itself might be different according to the design, including what loss is used. Once again, we thank the reviewer for this suggestion, and we have edited Section 6 accordingly.
>
> [1] Wu, Yiqing, et al. "Personalized prompt for sequential recommendation." IEEE Transactions on Knowledge and Data Engineering, 2024.

---

> > ### Author Response · Authors · 2024-11-24
> >
> > Dear reviewer, as the discussion period is ending soon, we sincerely hope to engage in more discussion. Thank you in advance!

---

### Official Review · Reviewer_QHJY · 2024-10-31

**Soundness:** 2
**Presentation:** 2
**Contribution:** 2
**Rating:** 3
**Confidence:** 3

**Summary:**

This paper studies the stability-plasticity trade-off for continual learning, with a focus on obtaining models for specified trade-off preferences. Differently from the typical rehearsal-based CL, which requires to retrain for every task and every preference, the authors propose a new paradigm named Imprecise Bayesian Continual Learning (UBCL) that replaces the retraining with weighted average of the extreme elements in the knowledge base, whose weights are provided by a preference vector. Since there is no additional training on the models, the entire procedure to achieve the provided trade-off is zero-shot. The main computation step is to compute the highest density region (HDR) under a distribution induced by weighted parameter posteriors of previous tasks. Empirically, the authors demonstrate that the proposed method outperforms existing algorithms in terms of accuracy significantly, with small overhead.

**Strengths:**

1.	The idea of using Bayesian techniques, FGCS and the convex combination of posteriors of previous tasks to achieve zero-shot training seems interesting in the context of continual learning.
2.	In the problems of continual learning under specific trade-offs (CluST), the proposed method significantly improves existing rehearsal-based and prompt-based algorithms.

**Weaknesses:**

1.	This paper is not well written. First, the motivation is not very clear. The paper considers CL under given preferences (i.e, weights). Although the paper gives some examples in recommendation systems, it does not talk much about how to get such weights. In addition, what is the setting of infinitely number of perferences? Some motivating examples are highly needed.
2.	In terms of the presentation, there are also multiple places to be clarified. First, in line 208, there is a probability $\hat q_{\bar w}$ that is not defined at the first place. I find its definition later in equation (1). I suggest using a different notation or explain clearly here. In algorithm 1, some details or explanations may be needed, since some readers may be not familiar with this procedure. In theorem 2, how to find the pareto-optimal parameters $\theta^*_{\bar w}$? Is there any guarantee that the output of the proposed algorithm is parato-optimal? More clarifications should be provided.
3.	The design contains multiple heuristics and multiple hyparameters to be deciede. For example, in algorithm 1, lines 4-8 remove similar elements, based on a threshold d. How to select d in practice? Is the performance sensitive to it? Why such elimination is useful? In algorithm 2, why choosing $\beta^1_k=….=\beta^m_k$? Is it just because of simple implementation? How about other choices? Finally, the number $m$ of distributions for each task is selected to be 3 in the experiments. How about other choices? Is there any trade-off between the accuracy and efficiency?
4.	The assumption made in this paper may be strong. In Assumption 1, it assumes all tasks have the same data distributions. It can happen that tasks share some level of similarity in data distribution, but they may involve different distribution components. In addition, in assumption 1, how to define $r$? This is because r can be either very large or very small. The final performance may be highly dependent on r. However, is there no such analysis.
5.	Experiments are not entire convincing. Algorithms are compared in term of accuracy but how about their performance in forgetting? In addition, perhaps I missed something, but are there any ablation studies on the selection of d, m, \alpha and other hyperparameters?

**Questions:**

Please refer to all the questions I mentioned in the weakness section. Overall, this paper gives an interesting idea but the proposed approach is not well explained and the results are not entire convincing. However, I am willing to raise my scores if my questions are well addressed.

---

> ### Author Response · Authors · 2024-11-19
>
> We appreciate the reviewer’s thoughtful comments. Possibly due to the presentation, we believe there are certain results already included in the paper but the reviewer has missed. We have edited the paper accordingly to improve the clarity. If any concern still persists, we are more than willing to discuss with the reviewer and edit our paper. We would be very grateful if the reviewer considers improving the rating.
>
> 1. Need to clarify the motivation.
>
> Our motivation is clarified in the second paragraph of Section 1. Specifically, the goal is to solve the CLuST problem efficiently by finding a method that generates every customized model per preference fast. This is because when there are a large number of preferences, if training each customized model is expensive, the overall cost will accumulate to a tremendous amount.
>
> 2. How to get preference weights?
>
> In this paper, we assume preference weights are given, which is the same assumption as in [1]. How to obtain preference weights shall be an interesting future research direction. One potential solution is to learn the preferences by sequential prompts [2]. We have added this point in Section 6.
>
> 3. What is a setting of infinite preferences?
>
> For infinite preferences, one example would be the movie recommendation system in our Section 1. As there may be a large number of customers, and each customer’s taste in movie genres may vary throughout time, there is potentially an infinite number of preferences. If the company has to train one model per preference, the cost would be huge. Therefore, a more efficient way of generating models to adapt to different preferences is needed, and IBCL is one solution. Generally, when there is a large number of users for a model to customize, IBCL has the advantage of efficient customization. We have edited Section 1 for clarity.
>
> 4. Need to clarify the presentation in multiple places.
>
> We thank the reviewer for pointing this out. We have made edits accordingly in the new version of paper. (1) We edited objective 2 in Section 3.2, by defining $\hat{q}_{\bar{w}}$ here. (2) We edited Algorithm 1, and added an explanation of variational inference after Algorithm 1 with a reference, for those who are unfamiliar with this procedure.
>
> 5. In Theorem 2, how to find Pareto-optimal parameters? Is there a guarantee?
>
> The Pareto-Optimal (PO) parameters are guaranteed to belong to the Highest Density Region that we build. Our algorithm does not find the PO parameter, but instead the narrowest region that contains it with high probability. In spirit, this result is very similar to what conformal prediction does (for predicted outputs, rather than parameters of interest). In practice, we can sample multiple parameters from the HDR, to estimate the Pareto-optimal parameters, as what we have done in the experiments — we sample 10 models per HDR and average them for estimation.
>
> 6. How is the performance in forgetting?
>
> We thank the reviewer for pointing out a potential issue in presentation clarity. We do measure forgetting metrics by backward transfer. These are in the every third subfigure in Figures 3, 4, 5 and 6, and discussed in the paragraph “As illustrated in the figures, IBCL has a slightly negative backward transfer in …” in Section 5.2. To make our presentation more clear, we have edited Section 5.1 on the setups.
>
> 7. Are there ablation studies on hyperparameters such as threshold $d$, and why choose equal $\beta$?
>
> In our edited version, we have ablation studies on threshold $d$, significance level $\alpha$, prior std sizes, numbers of priors, and $\beta$ in Appendix I. Equal $\beta$ is a choice of convenience, and the ablation studies show that it has the same performance as randomized $\beta$.
>
> (continue to the next comment)

---

> > ### Author Response · Authors · 2024-11-19
> >
> > (continued from the previous comment)
> >
> > 8. Why can we have Assumption 1?
> >
> > We thank the reviewer for giving us the opportunity of being clearer on this matter. Assumption 1 is much less strong than the reviewer points out. We do not assume that all tasks have the same distribution, but merely that the true data generating processes pertaining to different tasks are not too distant from one another. In addition, such a  notion of “being not too distant” is entirely in the hands of the user via the choice of radius $r$ and of the metric to endow the space $\Delta_{\mathcal{XY}}$ of distributions over $\mathcal{X}\times\mathcal{Y}$. We argue this is rather natural: for the time being, we do not expect, e.g., a robot to be able to fold our clothes (task 1), and then deliver a payload in a combat zone (task 2). As the reviewer correctly points out, “it can happen that tasks share some level of similarity in data distribution, but they may involve different distribution components”. This simply boils down to choosing the correct metric or divergence to define $\mathcal{F}$. In our work, we chose the 2-Wasserstein metric because of its ease of computing convex combinations. Furthermore, as the reviewer correctly points out, the diameter $r$ of $\mathcal{F}$ can impact the performance of IBCL. One future research direction is to study how varying the diameter of $\mathcal{F}$ impacts the performance of IBCL.
> >
> > [1] Mahapatra, Debabrata, and Vaibhav Rajan. "Multi-task learning with user preferences: Gradient descent with controlled ascent in pareto optimization." International Conference on Machine Learning. PMLR, 2020.
> >
> > [2] Wu, Yiqing, et al. "Personalized prompt for sequential recommendation." IEEE Transactions on Knowledge and Data Engineering, 2024.

---

> > > ### Author Response · Authors · 2024-11-24
> > >
> > > Dear reviewer, as the discussion period is ending soon, we sincerely hope to engage in more discussion. Thank you in advance!

---

> ### Comment · Reviewer_QHJY · 2024-11-25
> **Thanks for the response**
>
> I thank the authors for the response! Some of my questions have been resolved. However, I am not very satisfied about the answers to 2, 5 and 8. For 5, my question is about if there is any theoretical guarantee on the Pareto-optimality (for any solution or for the Highest Density Region). If yes, could I find any result in the paper. For 8, the assumption is still strong. It is still hard to guarantee that the true data generating processes pertaining to different tasks are not too distant from one another within a radius of $r$. For example, how to ensure and validate this assumption in practice? In many cases, the distributions over different tasks can be quite different from each other. What if $r$ is very large? How does it impact on the final performance?
>
> Best,
> Reviewer

---

> > ### Author Response · Authors · 2024-11-26
> >
> > Dear reviewer, thank you for your response. We address your concerns as follows.
> >
> > - Is there any theoretical guarantee on Pareto optimality?
> >
> > The guarantee we have is Theorem 2 in the paper. By construction, parameter $\theta^\star_{\bar{w}}$ parameterizes the Pareto-optimal distribution. Theorem 2 guarantees that, according to the distribution $\hat{q}_\bar{w}$
> >
> > (that is, according to the posterior that we obtain from the FGCS once we take into account the preference vector $\bar{w}$ over the different tasks), the parameter $\theta^\star_{\bar{w}}$ belongs to the HDR $\Theta^\alpha_{\bar{w}}$ that we derive in Algorithm 2, with prob. $\geq 1-\alpha$. In turn, by Assumption 2, this implies that, with $\hat{q}_\bar{w}$,
> >
> > we have probability at least $1-\alpha$, the Pareto-optimal distribution is parameterized by a parameter in the HDR $\Theta^\alpha_{\bar{w}}$. We will make this explicit in the final version, if there is a chance.
> >
> > - Assumption 1 is still strong.
> >
> > We agree that your concern makes sense, that in reality, it is hard to identify such a distance $r$. Still, this assumption is one standard assumption, i.e., task similarity, in continual learning, and we are not the first one using it. In fact, Assumption 1 expresses a bounded discrepancy in task distributions, as described in Figure 2, Section 3.2 of this survey paper [1]. How to validate and how to choose such a radius $r$ would be studied in a separate research. We are also more than willing to make this explicit in the final version.
> >
> > We will be very grateful if the reviewer can consider improving the ratings, given our responses and revised version of the paper.
> >
> > [1]  Wang, Liyuan, et al. "A comprehensive survey of continual learning: theory, method and application." IEEE Transactions on Pattern Analysis and Machine Intelligence (2024).

---

> > > ### Author Response · Authors · 2024-12-02
> > >
> > > Dear reviewer, as the extended discussion period is ending soon, we sincerely hope to engage in more discussion based on our latest response. Thank you in advance!

---

### Official Review · Reviewer_Ro7c · 2024-11-03

**Soundness:** 3
**Presentation:** 3
**Contribution:** 3
**Rating:** 8
**Confidence:** 2

**Summary:**

This paper models the problem of continual learning under specific trade-offs (CLuST) as a convex combination of previous tasks to the preference vector. The authors propose an algorithm named Imprecise Bayesian Continual Learning (IBCL) that transforms the convex combination of data distributions into the convex combination of model parameters under the framework of Bayesian learning. This algorithm is training-free for any newly arrived preference vectors compared to previous rehearsal-baed methods. The algorithm also performs well in numerical experiments.

**Strengths:**

1. The paper is written in a well-organized and self-contained way. The key concepts are clearly defined and sufficiently explained.
2. The idea of avoiding re-training models when receiving new preference vectors is smart. Transforming the convex combination of distributions into the convex combinations of posterior distributions of model parameters is natural, especially when the tasks are similar.
3. The numerical experiments verify the excellence of the algorithm. The code is also well-written.

**Weaknesses:**

1. The contribution is restricted to the domain-incremental continual learning scenario.
2. The algorithm's effectiveness relies on a core assumption that there is a continuous mapping from the data distribution to the distribution of the ground-truth model parameters and that mapping is (approximately) linear. The authors should specify this reliance and perhaps give more discussions on the validity of the assumption (for example, the dependence on the model/prior choices and the dependence on the underlying data distributions).
3. The theory part does not have an in-depth algorithm analysis. Theorem 1 is about the modeling rather than the algorithm. Theorem 2 is unnecessary: a) the theorem relies on the assumption that the Pareto-optimal parameter follows the estimated posterior distribution, which directly assumes the correctness of the estimation; b) the theorem does not provide any useful information since the coverage guarantee is already defined in the high-density region (HDR). I suggest deleting Theorem 2 to avoid confusion. Some more in-depth discussions are preferred.

**Questions:**

1. What are the benefits of IBCL compared to linearly weight interpolation? For example, for a preference vector, the linear combination of model weights $\sum_{j=1}^m q_j \hat{\theta}_j$ itself induces a model (here the $\hat{\theta}_j$ can be any estimated model weights, for example, via empirical risk minimization). Some theoretical analysis and numerical experiments would be preferred.

---

> ### Author Response · Authors · 2024-11-19
>
> We appreciate the reviewer for such a positive rating. Here we answer the question.
>
> What are the benefits of IBCL compared to linear weight interpolation?
>
> The major benefit is that using Bayesian models augments the search space. Say we have two models, parameterized by $\theta$ and $\theta’$, linear weight interpolation leads to a search space
>
> $\Theta(\theta, \theta') =$ {$\theta'' = w\theta + (1-w)\theta' \space | \space w \in [0, 1]$}
>
> In contrast, if we adopt Bayesian models $q = \mathcal{N}(\theta, \sigma^2)$ and $q’ = \mathcal{N}(\theta’, \sigma^2)$, where $\sigma$ is some selected std, we have
>
> $Q(q, q') =$ {$w q + (1-w) q' \space | \space w \in [0, 1]$}, and $\Theta_{aug}(\theta, \theta') = $ { $\theta \sim q_w \space | \space q_w \in Q(q, q')$}.
>
> Therefore, we have a chance to sample better models from $\Theta_{aug}$ than simply obtaining them from $\Theta$. This is also illustrated in our Figure 9 in Appendix I, where sampled models may be close to or far away from the Pareto front. Using Bayesian models, we can sample the ones closest to the Pareto front.

---

> > ### Comment · Reviewer_Ro7c · 2024-11-19
> >
> > Thanks for your response. I will maintain my positive score.

---

### Official Review · Reviewer_GJMR · 2024-11-04

**Soundness:** 2
**Presentation:** 1
**Contribution:** 3
**Rating:** 3
**Confidence:** 4

**Summary:**

The paper, introduces Imprecise Bayesian Continual Learning (IBCL) to address the Continual Learning under Specific Trade-offs (CLuST) problem. Traditional methods that balance stability and plasticity often rely on rehearsal-based learning, requiring retraining for each new trade-off, which is inefficient. IBCL offers a more efficient approach by constructing a convex hull of model parameter distributions and enabling zero-shot model generation for specific trade-offs without retraining. IBCL achieves this by updating a knowledge base in the form of a convex set of distributions and using convex combinations to generate Pareto-optimal models according to user-specified preferences. Experiments indicate that IBCL improves per-task accuracy and backward transfer compared to existing CLuST methods, with constant-time overhead and sub-linear memory growth.

**Strengths:**

1. The algorithm offers many favorable features, including the efficiency in model generation and sub-linear memory growth
2. It is innovative to investigate the problem through a Bayesian lens.

**Weaknesses:**

I do not think this paper is well written enough for me to follow easily. First, some definitions are not formal. For example Definition 1, 2 should be written more formally. See questions 1 and 2 below. Second, some important concepts should be presented in detail with formulae. For example, in line 218, continual Bayesian learning appears without a formal introduction. Third, the words and phrases should be picked more carefully. For example, in line 8 of Algorithm 1, one should state: store xxx and use xxx when xxx instead of saying remember xxx and use xxx later on.

**Questions:**

1. What is the definition of $q^j$? Is it a real vector?
2. In definition 2, what is $\int$ is a minimum?
3. Are $\mathcal{X}, \mathcal{Y}$ subsets of the Euclidean space?
4. Variational inference can be computationally intensive? Will this algorithm become computationally infeasible in real-world settings?

---

> ### Author Response · Authors · 2024-11-19
>
> We appreciate the reviewer for giving us the opportunity of being clearer. A revised version is uploaded per the concerns. Please let us know if anything still needs further clarification or formalization, and we are more than happy to edit them. We will be very grateful if the reviewer can consider improving the rating.
>
> 1. Some definitions are not formal, and wordings should be picked more carefully.
>
> We have edited Definition 1 and 2 in the paper, with a detailed explanation and illustrated example of Definition 2 in Appendix B. We also added Definition 3 to formalize Bayesian continual learning. Moreover, we edited Algorithm 1 and its discussion to make it more formal.
>
> 2. What is the definition of $q^j$?
>
> The $q^j$’s are probability distributions. Definition 1 says that, given a finite collection of distributions ${q^j}_{j=1}^m$, $\mathcal{Q}$ is its convex hull, that is, $\mathcal{Q}$ is the collection of all probability distributions that can be written as a convex combination of the $q^j$’s. As the reviewer intuitively points out, if the state space is finite then the $q^j$’s can indeed be seen as probability vectors, whose entries represent the probability mass assigned by distribution $q^j$ to the elements of the state space. We also point out how we used the canonical definition of convex hull (see e.g. [1]); we only slightly generalize it to be the convex hull of distributions instead of elements of a metric space.
>
> 3. In Definition 2, what is $\int$ is a minimum?
>
> Requiring that the integral is a minimum corresponds to requiring a minimal cardinality. Formally, this minimal integral generalizes the requirement of minimal cardinality to the case where the underlying set $\Theta$ may be uncountably infinite. In the continuous case, requiring that the integral is a minimum ensures us that the HDR is the set having the least amount of elements, which satisfies the desired condition. We also point out how we are not the first who introduce this notation; it was proposed first in [2]. A detailed definition with illustration can be found in Appendix B.
>
> 4. Are $\mathcal{X}$, $\mathcal{Y}$ a subset of Euclidean space?
>
> In a typical classification problem, $\mathcal{X}$ will be a subset of a Euclidean space, and $\mathcal{Y}$ a finite set. In a typical regression problem, $\mathcal{Y}$ will too be a subset of a Euclidean space. In general, we do not limit ourselves to either scenario. As a consequence, we purposefully let the input and the output spaces, $\mathcal{X}$ and $\mathcal{Y}$, respectively, as generic sets.
>
> 5. Why say variational inference is computational intensive? Is IBCL feasible in real-world scenarios?
>
> We say variational inference (VI) is computational intensive because it requires optimization of an objective function (usually ELBO loss). This is the major reason why state-of-the-art solutions are computationally expensive, as they must run one optimization per model generation. In contrast, IBCL (1) first runs a small fixed number of optimizations, and (2) then generates models via convex combination, which does not involve any optimization. This design makes IBCL more computationally feasible than state-of-the-art methods.
>
> [1] Phelps, Robert R., ed. "Lectures on Choquet’s theorem". Berlin, Heidelberg: Springer. 2001.
>
> [2] Coolen, Franciscus Petrus Antonius. "Imprecise highest density regions related to intervals of measures." 1992.

---

> > ### Author Response · Authors · 2024-11-24
> >
> > Dear reviewer, as the discussion period is ending soon, we sincerely hope to engage in more discussion. Thank you in advance!

---

> > ### Comment · Reviewer_GJMR · 2024-11-27
> >
> > After reading the revised version I feel that I can follow the paper well enough to understand the content. However I still feel there are places where this paper should be polished. Specifically, the technical content in the revised version is still not rigorously written enough. For example in Definition 1, the input space, or the support, of the distributions should be specified, at least in the Appendix. Especially for a paper trying to make theoretical contributions, stating the framework clearly and rigorously is very important.
> >
> > Regarding the contribution, I do not feel I learn anything interesting from reading this paper. In particular, I do gain new insight from formulating the problem theoretically as in the paper. I do not think the empirical result is surprising to me either. This might be my problem because I do not work in this area. I tried to read the related works of this paper, but I cannot find many recent publications (in three years) in top conferences like ICLR, so I am unable to know whether this paper is improving significantly. I would be happy if the authors clarify the major insight of this paper, or list some publications for me to compare.
> >
> > For now, I still maintain my score because of the writing issues and because I do not see anything particularly interesting about this work.

---

> > > ### Author Response · Authors · 2024-11-27
> > >
> > > We thank the reviewer for giving us a chance for further clarification. We address your concerns as follows.
> > >
> > > 1. "Stating the framework clearly and rigorously is important."
> > >
> > > We thank the reviewer for their remark. Definition 1 is the generic definition of an FGCS. That is, the support of the distributions $q^j$ is a generic measurable space of interest. It is defined similarly e.g. in [1]. We used the same notation that we later use for the parameter distributions because in this paper we are interested in building an FGCS of posterior parameter distributions, and we wanted to have notation continuity between the general definition (Definition 1), and the instance of FGCS that we consider in our CLuST problem. In the updated version of the paper, we will make it clear that the support is a generic (measurable) space $\Theta$ of interest, like we did for Definition 2. We are not sure what the reviewer refers to by "input space" of a distribution, but we assume they intend it as a synonym to "support". We are willing to make these explicit in our final version if there is a chance.
> > >
> > > 2. "I am unable to know whether the paper is improving significantly."
> > >
> > > We thank the reviewer for giving us a chance for further clarification, and we are more than willing to make it explicit in the final version, if there is a chance. The major contribution of this paper is that we are the first to formalize the problem of CLuST. For years, people in the area of multitask learning / continul learning have been working on balancing the performance of learning a new task (plasticity) and maintaining a low forgetting of previous tasks (stability), with well-cited publications in various venues, including ICLR [2, 3, 4, 5, 6]. Starting in the 2020s, researchers have been using quantitative preference vectors over learning tasks to balance the stability-plasticity trade-off [7, 8]. These preferences are used as weights to regularize loss functions on different tasks.
> > >
> > > Following this trend in continual learning research, we are the first to formalize the problem of CLuST, which specifies that the preference vectors are convex combination coefficients over task distributions, and the target combined distribution can be learned by convex combinations on Bayesian models corresponding to each task. This is a novel usage of preference vectors, which brings a huge efficiency advantage compared to using them as loss regularization weights.
> > >
> > > We are more than willing to engage in more discussion with the reviewer, and will be very grateful if the reviewer can consider improving the ratings accordingly.
> > >
> > > [1] Mauá, Denis Deratani, and Fabio Gagliardi Cozman. "Specifying credal sets with probabilistic answer set programming." International Symposium on Imprecise Probability: Theories and Applications. PMLR, 2023.
> > >
> > > [2] Kirkpatrick, James, et al. "Overcoming catastrophic forgetting in neural networks." Proceedings of the national academy of sciences 114.13 (2017): 3521-3526.
> > >
> > > [3] Kemker, Ronald, et al. "Measuring catastrophic forgetting in neural networks." Proceedings of the AAAI conference on artificial intelligence. Vol. 32. No. 1. 2018.
> > >
> > > [4] Serra, Joan, et al. "Overcoming catastrophic forgetting with hard attention to the task." International conference on machine learning (ICML). 2018.
> > >
> > > [5] Hayes, Tyler L., et al. "Remind your neural network to prevent catastrophic forgetting." European conference on computer vision (ECCV). 2020.
> > >
> > > [6] Ramasesh, Vinay Venkatesh, Aitor Lewkowycz, and Ethan Dyer. "Effect of scale on catastrophic forgetting in neural networks." International Conference on Learning Representations (ICLR). 2021.
> > >
> > > [7] Kim, Sanghwan, et al. "Achieving a better stability-plasticity trade-off via auxiliary networks in continual learning." Proceedings of the IEEE/CVF Conference on Computer Vision and Pattern Recognition (CVPR). 2023.
> > >
> > > [8] Mahapatra, Debabrata, and Vaibhav Rajan. "Multi-task learning with user preferences: Gradient descent with controlled ascent in pareto optimization." International Conference on Machine Learning (ICML). 2020.

---

> > > > ### Author Response · Authors · 2024-12-02
> > > >
> > > > Dear reviewer, as the extended discussion period is ending soon, we sincerely hope to engage in more discussion based on our latest response. Thank you in advance!

---

### Author Response · Authors · 2024-11-17
**Thank you for the reviews and a new version of paper is in progress**

Dear reviewers,

We deeply appreciate all your comments for giving us a chance to clarify things. We are working on addressing all the comments, including running additional ablation studies. A new version of the paper will be out soon. Thank you for your patience!

---

> ### Author Response · Authors · 2024-11-19
> **Revised version updated**
>
> Dear reviewers, thank you very much for your patience. We have uploaded our revised paper with additional experiments. Please refer to our comments, and we hope to engage in more discussions with all of you.

---

### Meta-Review · Area_Chair_Uz8y · 2024-12-22

**Metareview:**

The authors propose Imprecise Bayesian Continual Learning (IBCL) for zero-shot model generation under specified stability-plasticity trade-offs. Despite its innovative framing, there are several key weaknesses of this work. The writing is insufficiently rigorous, with unclear definitions and presentation gaps (e.g., vague assumptions and overly simplified algorithm descriptions). Critical theoretical guarantees (e.g., Pareto-optimality) are under-explored, and experimental validation lacks depth, especially regarding robustness to hyperparameter choices and task dissimilarities. While the idea has potential, these deficiencies limit its scientific contribution and practical applicability. Hence, I recommend rejection, as significant revisions and clarifications are needed before it reaches the standard for acceptance.

**Additional Comments On Reviewer Discussion:**

The discussion highlighted ongoing concerns regarding theoretical guarantees, assumptions about task similarity, and clarity in the mathematical framework. While the authors provided clarifications and edits, these responses did not fully address the reviewers' doubts. Reviewer QHJY maintained concerns about the validity of key assumptions and the limited practical applicability of results, while others noted a lack of novelty and rigorous analysis. Although some reviewers appreciated the idea’s potential, the lack of convincing empirical evidence and theoretical depth ultimately weighed against acceptance.

---

### Decision · Program_Chairs · 2025-01-22

Reject